# Learning (Approximately) Equivariant Networks via Constrained Optimization

**Andrei Manolache[124]**    **Luiz F.O. Chamon[3]**    **Mathias Niepert[12]**

[1]Computer Science Department, University of Stuttgart, Germany
[2]International Max Planck Research School for Intelligent Systems, Germany
[3]Department of Applied Mathematics, École Polytechnique de Paris, France
[4]Bitdefender, Romania
{andrei.manolache, mathias.niepert}@ki.uni-stuttgart.de
luiz.chamon@polytechnique.edu

## Abstract

Equivariant neural networks are designed to respect symmetries through their architecture, boosting generalization and sample efficiency when those symmetries are present in the data distribution. Real-world data, however, often departs from perfect symmetry because of noise, structural variation, measurement bias, or other symmetry-breaking effects. Strictly equivariant models may struggle to fit the data, while unconstrained models lack a principled way to leverage partial symmetries. Even when the data is fully symmetric, enforcing equivariance can hurt training by limiting the model to a restricted region of the parameter space. Guided by homotopy principles, where an optimization problem is solved by gradually transforming a simpler problem into a complex one, we introduce *Adaptive Constrained Equivariance* (ACE), a constrained optimization approach that starts with a flexible, non-equivariant model and gradually reduces its deviation from equivariance. This gradual tightening smooths training early on and settles the model at a data-driven equilibrium, balancing between equivariance and non-equivariance. Across multiple architectures and tasks, our method consistently improves performance metrics, sample efficiency, and robustness to input perturbations compared with strictly equivariant models and heuristic equivariance relaxations.

## 1   Introduction

Equivariant neural networks (NNs) [1–4] leverage known symmetries to improve sample efficiency and generalization, and are widely used in computer vision [1, 3, 5–7], graph learning [8–11], and 3D structure modeling [12–17]. Despite their advantages, training these networks can be challenging. Indeed, their inherent symmetries give rise to complex loss landscapes [18] even when the model matches the underlying data symmetry [18, 19]. Additionally, real-world datasets frequently include noise, measurement bias, and other symmetry-breaking effects such as dynamical phase transitions or polar fluids [20–24] that undermine the assumptions of strictly equivariant models [25–27]. These factors complicate the optimization of equivariant NNs, requiring careful hyperparameter tuning and often resulting in suboptimal performance.

A well-established strategy for approaching difficult optimization problems is to use *homotopy* (or continuation), i.e., to first solve a simple, surrogate problem and then track its solutions through a sequence of progressively harder problems until reaching the original one [28, 29]. Continuation has been used for compressive sensing [30–32] and other non-convex optimization problems [33–35], where it is closely related to simulated annealing strategies [36–40]. These approaches suggest that introducing equivariance gradually can facilitate training by guiding the model through a series

39th Conference on Neural Information Processing Systems (NeurIPS 2025).

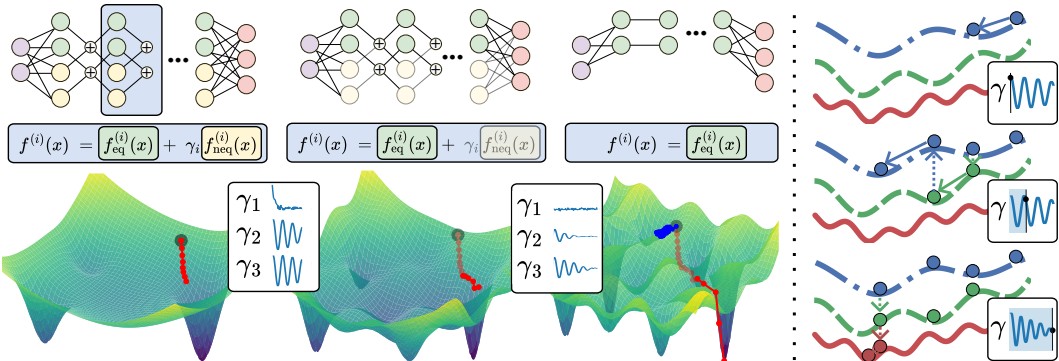

Figure 1: The proposed ACE homotopic optimization scheme at a glance. **Left:** The red trajectory illustrates the training of a relaxed, non-equivariant model that gradually becomes equivariant as the layer-wise coefficients $\gamma_i$ decay. The blue trajectory illustrates a strictly equivariant network $f_{eq}$ trained from the same initialization. **Right:** the coefficient $\gamma$ allows the training problem to transition between easier (non-equivariant) and harder (equivariant) problems, facilitating convergence.

of less stringent intermediate optimization problems. However, implementations of this strategy based on modifying the training objective [18] or the underlying model [41] require domain-specific knowledge to design the intermediate stages and craft loss penalties, limiting practical usability.

This work, therefore, addresses the following question: *is there an automated, general-purpose strategy to navigate these intermediate problems that precludes manual tuning and that works effectively for various models and tasks?* To this end, we leverage the connection between homotopy approaches and dual methods in optimization, which both provide rigorous frameworks for solving constrained problems by addressing a sequence of relaxed problems with progressively stricter penalties. By casting the training process as a constrained optimization problem, we introduce *Adaptive Constrained Equivariance* (ACE), a principled framework for automatically relaxing/tightening equivariance in NNs. Our method balances downstream performance (e.g., accuracy) with equivariance, relaxing equivariance at initial stages while gradually transitioning to a strictly equivariant model, as illustrated in Fig. 1. This transition occurs in an automated, data-driven fashion, eliminating much of the trial-and-error associated with manual tuning.

We summarize our contributions as follows:

1. **A general equivariant training framework.** We introduce ACE, a principled constrained optimization method to train equivariant models by automatically transitioning in a data-driven fashion from a flexible, non-equivariant model to one that respects symmetries without manually tuned penalties, weights, or schedules (Sec. 4).

2. **Accommodating partial equivariance.** Our data-driven method can identify symmetry-breaking effects in the data and automatically relax equivariance to mitigate their impact on downstream performance while exploiting the remaining partial equivariance (Sec. 4.2).

3. **Theoretical guarantees and empirical alignment.** We provide explicit bounds on the approximation error and equivariance level of the resulting model (Thm. 4.1–4.2) and show how they are reflected in practice.

4. **Comprehensive empirical evaluation.** We conduct a systematic study across four representative domains to examine the impact of our algorithm on convergence, sample efficiency, and robustness to input degradation compared to strictly equivariant and relaxed alternatives.

## 2   Related Work

Equivariant NNs encode group symmetries directly in their architecture. In computer vision, for instance, equivariant convolutional NNs accomplish this by tying filter weights for different rotations and reflections [1–4]. Equivariant NNs have also been extended to 3D graphs and molecular systems, as is the case of SchNet, that operates directly on atomic coordinates while remain-

ing $\mathsf{E}(3)$-invariant [12], and Tensor Field Networks or $\mathsf{SE}(3)$ Transformers, that extend this to full roto-translation equivariance using spherical harmonics and attention, respectively [13, 42].

Scalable, parameter-efficient architectures that remain equivariant include EGNNs [43], SEG-NNs [44], and methods such as Vector Neurons [45], which lift scalars to 3D vectors, achieving exact $\mathsf{SO}(3)$ equivariance in common layers with minimal extra parameters. Moreover, recent works model 3D trajectories rather than static snapshots, such as EGNO, which uses $\mathsf{SE}(3)$ equivariant layers combined with Fourier temporal convolutions to predict spatial trajectories [46].

While exact equivariance can improve sample efficiency [47–49], it may render the optimization harder, particularly when the data violates symmetry (even if partially) due to noise or other symmetry-breaking phenomena [22–24]. This has motivated the use of models with *approximate* or *relaxed* equivariance based on architectures that soften weight-sharing, such as Residual Pathway Priors [50] that adds a non-equivariant branch alongside the equivariant one, or works such as Wang et al. [26], where weighted filter banks are used to interpolate between equivariant kernels. Recent works have formalized the trade-offs that appear due to these symmetry violations, establishing generalization bounds that specify when and how much equivariance should be relaxed [19].

Beyond architectural modifications, another line of work relaxes symmetry *during training*. REMUL [18] adds a penalty for equivariance violations to the training objective, adapting the penalty weights throughout training to balance accuracy and equivariance. This procedure does not guarantee that the final solution is equivariant or even the degree of symmetry achieved. An alternative approach modifies the underlying architecture by perturbing each equivariant layer and gradually reducing the perturbation during training according to a user-defined schedule [41]. This guided relaxation facilitates training and guarantees that the final solution is equivariant, but is sensitive to the schedule. Additional equivariance penalties based on Lie derivative are used to mitigate this issue, increasing the number of hyperparameters and restricting the generality of the method.

The method proposed in this paper addresses many of the challenges in these exact and relaxed equivariant NN training approaches, namely, (i) the complex loss landscape induced by equivariant NNs; (ii) the manual tuning of penalties, weights, and schedules; (iii) the need to account for partial symmetries in the data.

## 3   Training Equivariant Models

Equivariant NNs incorporate symmetries inherent to the data by ensuring that specific transformations of the input lead to predictable transformations of the output. Formally, let $f_\theta : X \to Y$ be a NN parameterized by $\theta \in \mathbb{R}^p$, the network weights. A NN is said to be *equivariant* to a group $G$ with respect to actions $\rho_X$ on $X$ and $\rho_Y$ on $Y$ if

$$f_\theta\big(\rho_X(g)x\big) = \rho_Y(g)f_\theta(x), \quad \text{for all } x \in X \text{ and } g \in G. \tag{1}$$

Equivariant NNs improve sample efficiency by encoding transformations from a symmetry group $G$ [47–49]. Strictly enforcing equivariance, however, limits the expressiveness of the model and create intricate loss landscapes, which can slow down optimization and hinder their training [19, 18], particularly in tasks involving partial or imperfect symmetries [20–24].

To temper these trade-offs, recent work relaxes symmetry during training by modifying either the optimization objective or the model itself [41, 18, 50, 26, 19]. Explicitly, let $f_{\theta,\gamma} : X \to Y$ be a NN now parameterized by both $\theta$, the network weights, and $\gamma \in \mathbb{R}^L$, a set of *homotopic parameters* representing the model *non-equivariance*. The model is therefore equivariant for $\gamma = 0$, i.e., $f_{\theta,0}$ satisfies (1), but can behave arbitrarily if $|\gamma_i| > 0$ for any $i$. Consider training this NN by solving

$$\underset{\theta}{\text{minimize}} \ \alpha\left[\frac{1}{N}\sum_{n=1}^{N}\ell_0\big(f_{\theta,\gamma}(x_n), y_n\big)\right] + \beta\ell_{\text{eq}}\big(f_{\theta,\gamma}\big), \tag{2}$$

where $(x_n, y_n)$ are samples from a data distribution $\mathfrak{D}$, $\ell_0$ denotes the top-line objective function (typically, mean-squared error or cross-entropy), $\ell_{\text{eq}}$ denotes an equivariance metric, and $\alpha, \beta > 0$ are hyperparameters controlling their contributions to the training loss. Hence, traditional equivariant NN training amounts to using $\alpha = 1$ and $\beta = \gamma = 0$ in (2). In [18], a non-equivariant model ($\gamma = 1$) is trained using $\ell_{\text{eq}}(f) = \sum_n \mathbb{E}_{g_n}\big[\|f\big(\rho_X(g_n)x_n\big) - \rho_Y(g_n)f(x_n)\|^2\big]$ with $g_n$ sampled uniformly

from $G$. The values of $\alpha, \beta$ are adapted to balance the contributions of each loss, i.e., to balance accuracy and equivariance. In contrast, [41] uses an equivariance metric $\ell_{\text{eq}}$ based on Lie derivatives [51, 52], fix $\alpha, \beta$, and manually decrease $\gamma$ throughout training until it vanishes, i.e., until the model is equivariant.

While effective in particular instances, these approaches are based on sensitive hyperparameters, some of which are application-dependent. This is particularly critical in [18], where both performance and equivariance depend on the values of $\alpha, \beta$. While [41] guarantees equivariance by driving $\gamma$ to zero, its performance is substantially impacted by the rate at which it vanishes. It therefore continues to use $\ell_{\text{eq}}$ even though it is not needed to ensure equivariance. What is more, it is once again susceptible to symmetry-breaking effects in the data since it strictly imposes equivariance. In the sequel, we address these challenges by doing away with hand-designed penalties ($\beta = 0$) *and* manually-tuned schedules for $\gamma$. Before proceeding, however, we briefly comment on the implementation of $f_{\theta,\gamma}$.

**Homotopic architectures.** Throughout this paper, we consider an architecture similar to [41] built by composing equivariant layers linearly perturbed by non-equivariant components, e.g., a linear layer or a small multilayer perceptron (MLP). Formally, let

$$f_{\theta,\gamma} = f_{\theta,\gamma}^L \circ \cdots \circ f_{\theta,\gamma}^1 \quad \text{with} \quad f_{\theta,\gamma}^i = f_\theta^{\text{eq},i} + \gamma_i f_\theta^{\text{neq},i}, \quad i = 1, \ldots, L, \tag{3}$$

where $\gamma = (\gamma_1, \ldots, \gamma_L)$; $f^i : Z_{i-1} \to Z_i$ (and thus similarly for $f^{\text{eq},i}$ and $f^{\text{neq},i}$), with $Z_0 = X$ and $Z_L = Y$; and $f^{\text{eq},i}$ equivariant with respect to the actions $\rho_i$ of a group $G$ on $Z_i$, i.e., $f^{\text{eq},i}\big(\rho_{i-1}(g)z\big) = \rho_i(g)f^{\text{eq},i}(z)$ for all $g \in G$. Note that $\rho_0 = \rho_X$ and $\rho_L = \rho_Y$. We generally take $Z_i = \mathbb{R}^{k_i}$ and the group action $\rho_i$ to be a map from $G$ to the general linear group $\text{GL}_{k_i}$ (i.e., the group of invertible $k_i \times k_i$ matrices). Note that although we use the same set of parameters $\theta$ for all layers, they need not share weights and can use disjoint subsets of $\theta$. Immediately, the NN is equivariant for $\gamma = 0$, since it is a composition of equivariant functions $f^{\text{eq}}$. Otherwise, it is allowed to deviate from equivariance by an amount controlled by the magnitude of $\gamma$. It is worth noting, however, that the methods we put forward in the sequel do not rely on this architecture and apply to any model $f_{\theta,\gamma}$ that is differentiable with respect to $\theta$ and $\gamma$ and such that $f_{\theta,0}$ is equivariant.

## 4 Homotopic Equivariant Training

In this section, we address the challenges of equivariant training and previous relaxation methods, namely (i) the complex loss landscape induced by equivariant NNs ($\gamma = 0$); (ii) the manual tuning of penalties, weights, and/or homotopy schedules; (iii) the detection and mitigation of partial symmetries in the data. We use the insight that dual methods in constrained optimization are closely related to homotopy or simulated annealing in the sense that they rely on a sequence of progressively more penalized problems. This suggests that (i)–(ii) can be addressed in a principled way by formulating equivariant NN training as a constrained optimization problem (Sec. 4.1). We can then analyze the sensitivity of dual variables to detect the presence of symmetry violations in the data and relax the equivariance requirements on the final model [Sec. 4.2, (iii)]. We also characterize the approximation error and equivariance violations of the homotopic architecture (3) (Thm. 4.1–4.2).

### 4.1 Strictly equivariant data: Equality constraints

We begin by considering the task of training a strictly equivariant NN, i.e., solving (2) with $\alpha = 1$ and $\beta = \gamma = 0$. Note that this task is equivalent to solving

$$
\begin{aligned}
\underset{\theta,\gamma}{\text{minimize}} \quad & \mathbb{E}_{(x,y)\sim\mathfrak{D}}\Big[\ell_0\big(f_{\theta,\gamma}(x), y\big)\Big] \\
\text{subject to} \quad & \gamma_i = 0, \qquad \text{for } i = 1, \ldots, L.
\end{aligned}
\tag{PI}
$$

Indeed, only equivariant models $f_{\theta,0}$ are feasible for (PI), so that its solution must be equivariant. While (PI) may appear to be a distinction without a difference, its advantages become clear when we consider solving it using duality. Specifically, when we consider its empirical dual problem

$$\underset{\lambda \in \mathbb{R}^L}{\text{maximize}} \ \underset{\theta,\gamma}{\min} \ \hat{L}(\theta, \gamma, \lambda) \triangleq \frac{1}{N} \sum_{n=1}^N \ell_0\big(f_{\theta,\gamma}(x_n), y_n\big) + \sum_{i=1}^L \lambda_i \gamma_i, \tag{DI}$$

| **Algorithm 1** Strictly equivariant data | **Algorithm 2** Partially equivariant data |
|---|---|
| 1: **Inputs**: $\eta_p, \eta_d > 0$, $\gamma^{(0)} = 1$, $\lambda^{(0)} = 0$ | 1: **Inputs**: $\eta_p, \eta_d > 0$, $\gamma^{(0)} = 1$, $\lambda^{(0)} = 0$ |
| 2: $J_0^{(t)} = \dfrac{1}{N}\sum_{n=1}^{N} \ell_0\big(f_{\theta^{(t)},\gamma^{(t)}}(x_n), y_n\big)$ | 2: $J_0^{(t)} = \dfrac{1}{N}\sum_{n=1}^{N} \ell_0\big(f_{\theta^{(t)},\gamma^{(t)}}(x_n), y_n\big)$ |
| 3: $\theta^{(t+1)} = \theta^{(t)} - \eta_p \nabla_\theta J_0^{(t)}$ | 3: $\theta^{(t+1)} = \theta^{(t)} - \eta_p \nabla_\theta J_0^{(t)}$ |
| 4: $\gamma_i^{(t+1)} = \gamma_i^{(t)} - \eta_p\Big(\nabla_{\gamma_i} J_0^{(t)} + \lambda_i^{(t)}\Big)$ | 4: $\gamma_i^{(t+1)} = \gamma_i^{(t)} - \eta_p\Big(\nabla_{\gamma_i} J_0^{(t)} + \lambda_i^{(t)}\Big)$ |
| | 5: $u_i^{(t+1)} = u_i^{(t)} + \eta_p(\rho u_i^{(t)} - \lambda_i^{(t)})$ |
| 5: $\lambda_i^{(t+1)} = \lambda_i^{(t)} + \eta_d \gamma_i^{(t)}$ | 6: $\bar{\lambda}_i^{(t+1)} = \Big[\lambda_i^{(t)} + \eta_d(|\gamma_i^{(t)}| - u_i^{(t)})\Big]_+$ |

where $(x_n, y_n)$ are i.i.d. samples from $\mathfrak{D}$, $\lambda \in \mathbb{R}^L$ collects the *dual variables* [53, 54] $\lambda_i$, and $\hat{L}$ is called the *empirical Lagrangian* [55, 56]. In general, solutions of (PI) and (DI) are not related due to (i) non-convexity, which hinders strong duality [57], and (ii) the empirical approximation of the expectation in (PI) by samples. Yet, for rich enough parametrizations $f_{\theta,\gamma}$, constrained learning theory shows that solutions of (DI) do generalize to those of (PI) [53–55]. In other words, (DI) approximates the solution of the equivariant NN training problem (PI) without constraining the model to be equivariant.

More precisely, (DI) gives rise to a gradient descent (on $\theta, \gamma$) and ascent (on $\lambda$) algorithm (Alg. 1) that acts as an annealing mechanism adapted to the downstream task, taking into account the effect of enforcing equivariance on the objective function. Indeed, by taking $\gamma_i^{(0)} = 1$, Alg. 1 starts by training a flexible, non-equivariant model (step 3). The gradient step on the homotopy parameters $\gamma_i$ (step 4) then simultaneously seeks to reduce the objective function $\ell_0$ and bring $\gamma_i$ closer to zero, targeting the constraint in (PI). The strength of that bias toward zero (namely, $\lambda_i$) depends on the accumulated constraint violations (step 5): the longer $\gamma_i$ is strictly positive (negative), the larger the magnitude of $\lambda_i$. Hence, the $\gamma_i$ can increase to expand the flexibility of the model or decrease to exploit the symmetries in the data. The initial non-equivariant NN is therefore turned into an equivariant one by explicitly taking into account the effects this has on the downstream performance, instead of relying on hand-tuned, *ad hoc* penalties or schedules as in [18, 41].

Naturally, the iterates $\gamma_i^{(t)}$ do not vanish in any finite number of iterations. In fact, the behavior of saddle-point algorithms such as Alg. 1 are intricate even in convex settings [58, 59] and its iterates may, even for small learning rates $\eta_p, \eta_d$, (i) asymptotically approach zero or (ii) display oscillations that do not subside. In case (i), if we stop Alg. 1 at iteration $T$ and deploy its solution with $\gamma = 0$ (i.e., $f_{\theta^{(T)},0}$), we introduce an error that depends on the magnitude of $\gamma_i^{(T)}$. The following theorem bounds this approximation error for the architecture in (3) under mild assumptions on its components. In particular, note that Ass. 1 holds, e.g., for MLPs with ReLU activations and weight matrices of bounded spectral norms [60].

**Assumption 1.** The architecture in (3) is such that, for all $\theta$ and $i$, $f_\theta^{\mathrm{eq},i}$ and $f_\theta^{\mathrm{neq},i}$ are $M$-Lipschitz continuous and $f_\theta^{\mathrm{neq},i}$ is a bounded operator, i.e., $\|f_\theta^{\mathrm{neq},i}(x)\| \leq B\|x\|$ for all $x \in X$.

**Theorem 4.1.** *Consider $f_{\theta,\gamma}$ as in (3) satisfying Assumption 1. If $\bar{\gamma} \triangleq \max_i |\gamma_i|$, then*

$$\big\| f_{\theta,\gamma}(x) - f_{\theta,0}(x) \big\| \leq \left[\sum_{k=0}^{L-1} (1+\bar{\gamma})^k\right] \bar{\gamma} B M^{L-1} \|x\|, \quad \text{for all } x \in X.$$

A proof of this as well as a more refined bound is available in the Appendix A.6. Theorem 4.1 shows that as long as the $\{\gamma_i^{(T)}\}$ are small enough, the error incurred from setting them to zero is negligible. This is not the case, however, when the $\gamma_i$ oscillate without subsiding [case (ii)]. As we argue next, this effect can be used to identify and correct for symmetry deviations in the data.

## 4.2 Partially equivariant data: Resilient constraints

In some practical scenarios, the data used for training may only partially satisfy the equivariance relation (1) due to the presence of noise, measurement bias, or other symmetry-breaking effects [20–27]. In such cases, the equivariance constraints in (PI) become too stringent and the gradient of the objective with respect to $\gamma_i$ in step 4 of Alg. 1 does not vanish as $\gamma_i$ approaches zero. Alg. 1 will then oscillate as steps 4 and 5 push $\gamma_i$ alternatingly closer and further from zero [case (ii)] or $\gamma_i$ will settle, but $\lambda_i$ will not vanish. These observations can then be used as evidences that the data does not fully adhere to imposed symmetries. This is not necessarily a sign that equivariance should be altogether abandoned, but that we can improve performance by relaxing it. To this end, we rely on the fact that $\gamma$ provides a measure of non-equivariance, which we formalize next.

**Theorem 4.2.** *Consider $f_{\theta,\gamma}$ as in (3) satisfying Assumption 1 and suppose that the group actions are bounded operators, i.e., for all $i$ it holds that $\|\rho_i(g)z\| \leq B\|z\|$ for all $g \in G$ and $z \in Z_i$. If $\bar{\gamma} \triangleq \max_i |\gamma_i|$ and $C \triangleq \max(B,1)$, then*

$$\big\|\rho_Y(g)f_{\theta,\gamma}(x) - f_{\theta,\gamma}\big(\rho_X(g)x\big)\big\| \leq 2\bar{\gamma}\left(M + C\bar{\gamma}\right)^{L-1} LB^2 \|x\|, \quad \text{for all } x \in \mathcal{X} \text{ and } g \in G.$$

Once again, the proof of a more refined bound can be found in Appendix A.6. Also note that the bounded assumption on the group action is mild and that in many cases of interest the group action is in fact isometric, i.e., $\|\rho_i(g)z\| = \|z\|$ (e.g., rotation, translation). Theorem 4.2 shows that by adjusting the degree to which the equivariance constraint in (PI) is relaxed, we can control the extent to which the final solution remains equivariant. While we could replace the equality in (PI) by $|\gamma_i| \leq u_i$ for $u_i \geq 0$, choosing a suitable set of $\{u_i\}$ is not trivial, since they must capture how much partial equivariance we expect or can tolerate at each layer. Setting the $u_i$ too strictly degrades downstream performance (objective function $\ell_0$), whereas setting them loosely interferes with the ability of the model to exploit whatever partial symmetry is in the data. Striking this balance manually is impractical, especially as the number of layers (and consequently $\gamma_i$) increases.

Instead, we use the outcomes of Alg. 1 to determine which constraints in (PI) to relax and by how much. Indeed, notice that we only need to relax those $\gamma_i$ that do not vanish, seen as they are the ones too strict for the data. We can then leverage the fact that stricter constraints lead to larger $\lambda_i$ [53, 57] and relax each layer proportionally to their corresponding dual variable. In doing so, we reduce the number of $\gamma_i \neq 0$ as well as their magnitude, thus improving downstream performance (as measured by $\ell_0$) while preserving equivariance as much as possible (Theorem 4.2).

This trade-off rationale can be formalized and achieved automatically without repeatedly solving Alg. 1 by using *resilient constrained learning* [53, 54, 61]. Concretely, we replace (PI) by

$$\begin{aligned}
\underset{\theta,\gamma,u}{\text{minimize}} \quad & \mathbb{E}_{(x,y)\sim\mathfrak{D}}\left[\ell_0\big(f_{\theta,\gamma}(x), y\big)\right] + \frac{\rho}{2}\|u\|^2 \\
\text{subject to} \quad & |\gamma_i| \leq u_i, \qquad \text{for } i = 1, \ldots, L,
\end{aligned} \tag{PII}$$

where $u \in \mathbb{R}_+^L$ collects the *slacks* $u_i$ and $\rho > 0$ is a *proportionality constant* (we use $\rho = 1$ throughout). Notice that the slacks are not hyperparameters, but optimization variables that explicitly trade off between downstream performance, that improves by relaxing the constraints (i.e., increasing $u_i$), and constraint violation (i.e., the level of equivariance). In fact, it can be shown that a solution of (PII) satisfies $u_i^\star = \lambda_i^\star/\rho$, where $\lambda_i^\star$ is the Lagrange multiplier of the corresponding constraint [53, Prop. 3]. Using duality, (PII) lead to Alg. 2, where $[z]_+ = \max(z, 0)$ denotes a projection onto the non-negative orthant. Notice that it follows the same steps 3 and 4 of Alg. 1. However, it now contains an additional gradient descent update for the slacks (step 5) and the dual variable updates (step 6) now has a projection due to the constraints in (PII) being inequalities [57]. During training, Alg. 2 therefore automatically balances the level of equivariance imposed on the model ($u_i$) and the relative difficulty of imposing this equivariance ($\lambda_i$) as opposed to improving downstream performance ($J_0$).

## 5 Experimental Setup and Results

In this section, we aim to answer the following three research questions:

  **RQ1.** *Final Performance*: How does gradually imposing equivariance via ACE affect downstream task accuracy and sample efficiency across diverse domains and models?

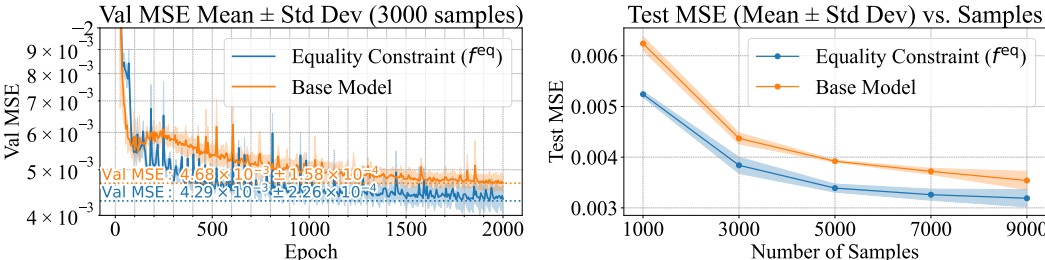

Figure 2: SEGNN trained with ACE equality constraints compared with the normal SEGNN on the N-Body dataset. **Left:** Validation MSE over 2000 epochs. **Right:** Test MSE versus training set size.

**RQ2.** *Training Dynamics and Robustness:* How do ACE constrained equivariance strategies, namely equality constrained optimization and partial equivariance enforced via resilient inequality constraints, affect convergence under input degradation, guide predictive performance when relaxing strict equivariance, and reveal where flexibility is most beneficial through their learned parameters?

**RQ3.** *Theory vs. Empirics:* To what extent do the theoretical equivariance-error bounds (Theorem 4.2) predict the actual deviations observed during training?

## 5.1 Final Performance and Sample Efficiency

**N-Body Simulations.** To address **RQ1**, we begin with the N-Body simulations dataset [63]. In Figure 2 (left), we compare the validation MSE (mean±std) across 2000 epochs for the "vanilla" SEGNN and an equivariant projection $f^{eq}$ obtained by training with ACE. Remarkably, $f^{eq}$ trained with ACE converges more rapidly and attains a lower final MSE than the vanilla baseline, demonstrating both accelerated training dynamics and greater stability. Figure 2 (right) shows test MSE with respect to the training set size. Here, our $f^{eq}$ trained on only 5000 samples matches (and slightly exceeds) the vanilla SEGNN's performance at 9000 samples, corresponding to a $\approx 44\%$ reduction in required data. Table 2 (left) presents test MSE results ($\times 10^{-3}$) for various SEGNN variants and baselines. Our SEGNN trained with ACE, $f^{eq}$, outperforms all other baselines, including the heuristic-based relaxed equivariance scheme of Pertigkiozoglou et al. [41] and, after projection, matches the performance of a non-equivariant SEGNN that retains the same architecture $f^{eq} + f^{neq}$ but, without using ACE, relies on explicit $SO(3)$ augmentations to learn equivariance.

**Molecular Property Regression.** We evaluate on the QM9 dataset [64, 65] using the invariant SchNet [12] and our ACE variant. Due to its strict invariance, SchNet may severely hinder convergence without any relaxation; indeed, as shown in Table 2 (right), incorporating ACE during training reduces the mean absolute error (MAE) across all quantum-chemical targets. To assess the impact of ACE on a more flexible architecture, we compare a standard SEGNN [44] against its ACE counterpart on QM9. As shown in Table 2 (right), the ACE SEGNN lowers the MAE on most targets; however, the overall performance remains largely comparable. Collectively, these findings show that SEGNN's E(3)-equivariance aligns well with the QM9 dataset, facilitating optimization. However, ACE can still provide an additional boost in performance for certain molecular properties.

**3D Shape Classification.** On the ModelNet40 classification benchmark [66], we apply ACE to the VN-DGCNN architecture [67, 45]. Figure 7 (left) shows that our ACE equivariant model $f^{eq}$ achieves gains of $+2.78$ and $+1.77$ percentage points for the class and instance accuracy metrics over normal training. The partially equivariant ACE variant $f^{eq} + f^{neq}$ achieves comparable improvements. Both models are obtained by checkpointing the same training runs.

## 5.2 Training Dynamics and Robustness

**Noisy 3D Shape Classification.** To address **RQ2**, we first evaluate convergence under input degradation on ModelNet40 [66] by randomly dropping $0\text{-}85\%$ of the points in the 3D point clouds of every shape. As shown in Figure 7 (right), the strictly equivariant baseline fails to converge under this

Equivariant

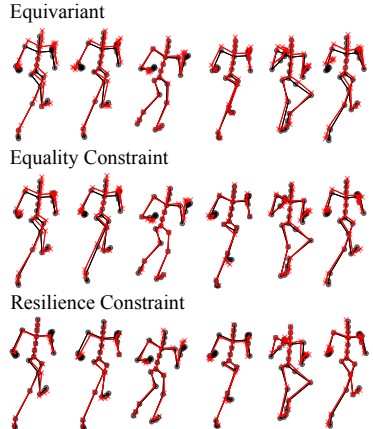

Equality Constraint

Resilience Constraint

Figure 3: Qualitative comparison on the CMU MoCap dataset (Subject #9, Run). **Top row:** Standard equivariant EGNO. **Middle row:** Partially equivariant EGNO ($f^{\mathrm{eq}} + f^{\mathrm{neq}}$) trained with an ACE equality constraint. **Bottom row:** Partially equivariant EGNO ($f^{\mathrm{eq}} + f^{\mathrm{neq}}$) trained with an ACE resilient inequality constraint, yielding significant improvements over both.

Table 1: Test MSE ($\times 10^{-2}$) on the CMU MoCap dataset for Subject #9 (Run) and Subject #35 (Walk). Results are shown for the standard equivariant models, EGNO models trained with an equality constraint (ACE†) and an EGNO model trained with an resilient inequality constraint (ACE‡). The partially equivariant EGNO ($f^{\mathrm{eq}} + f^{\mathrm{neq}}$) outperforms both the standard and equality-constrained projection $f^{\mathrm{eq}}$ models, with the adaptive-resilience variant achieving the lowest error on both subjects.

| Model | MSE↓ (Run) | MSE↓ (Walk) |
|---|---|---|
| MPNN [9] | $66.4_{\pm 2.2}$ | $36.1_{\pm 1.5}$ |
| RF [62] | $521.3_{\pm 2.3}$ | $188.0_{\pm 1.9}$ |
| TFN [13] | $56.6_{\pm 1.7}$ | $32.0_{\pm 1.8}$ |
| SE(3)-Tr.[42] | $61.2_{\pm 2.3}$ | $31.5_{\pm 2.1}$ |
| EGNN [43] | $50.9_{\pm 0.9}$ | $28.7_{\pm 1.6}$ |
| EGNO (report.) [46] | $\mathbf{33.9}_{\pm 1.7}$ | $\mathbf{8.1}_{\pm 1.6}$ |
| EGNO (reprod.) [46] | $\mathbf{35.3}_{\pm 3.2}$ | $8.5_{\pm 1.0}$ |
| EGNO$^{\mathrm{ACE†}}_{f_{\mathrm{eq}}}$ | $\mathbf{35.3}_{\pm 1.6}$ | $\mathbf{7.9}_{\pm 0.3}$ |
| EGNO$^{\mathrm{ACE†}}_{f^{\mathrm{eq}}+f^{\mathrm{neq}}}$ | $\mathbf{32.6}_{\pm 1.6}$ | $\mathbf{7.5}_{\pm 0.3}$ |
| EGNO$^{\mathrm{ACE‡}}_{f^{\mathrm{eq}}+f^{\mathrm{neq}}}$ | $\mathbf{23.8}_{\pm 1.5}$ | $\mathbf{7.4}_{\pm 0.2}$ |

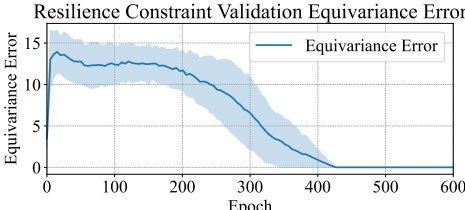

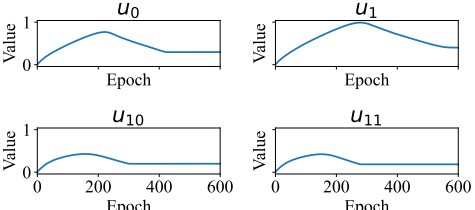

Figure 4: Validation equivariance error during training of a partially equivariant EGNO model with an ACE resilient inequality constraint on the CMU MoCap dataset (Subject #9, Run). The equivariance error decreases and approaches values near zero.

Figure 5: Adaptive constraint values $u$ on the first two and last two layers of an EGNO model trained with ACE resilient inequality constraints on the CMU MoCap dataset (Subject #9, Run). Early layers allow large slacks that decrease over training, while later layers stay more tightly constrained.

perturbation, whereas our equality-constrained projection $f^{\mathrm{eq}}$ converges quickly to a substantially higher test accuracy.

**Relaxed Equivariance for Motion Capture.** Next, we evaluate predictive performance on the CMU Motion Capture (MoCap) "Run" and "Walk" datasets [69], where we examine the effect of training with both the equality constraint and the inequality constraint of resilient constrained learning. As shown in Table 1, the equivariant projection $f^{\mathrm{eq}}$ performs similar to the vanilla EGNO. However, the partially equivariant $f^{\mathrm{eq}} + f^{\mathrm{neq}}$ outperforms both models, indicating that strictly equivariant models might not align well with the dataset. To examine whether better predictive performance can be obtained with a partially equivariant model, we perform experiments with resilient inequality constraints and observe that this training strategy further reduces test MSE on both *Run* and *Walk* compared to the purely equivariant model and the ones trained with equality constraints. These results confirm that resilience parameters can adaptively allocate non-equivariant capacity where it is most needed, improving overall performance.

**Modulation Coefficients Dynamics.** To better understand the effects of ACE, we inspect how the constrained parameters evolve during training. On ModelNet40, the modulation coefficients $\gamma_k$ in the equality-constrained VN-DGCNN decay rapidly and reach zero within the first third

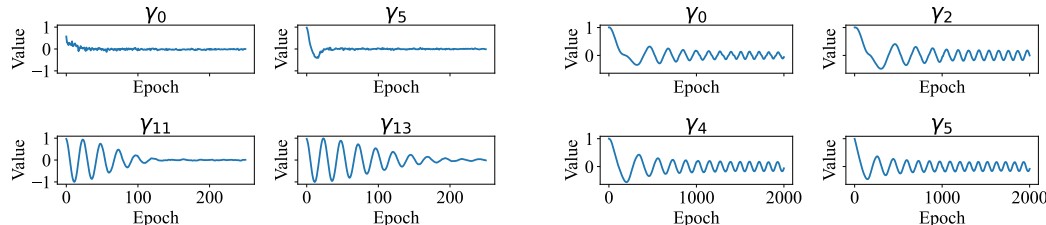

Figure 6: Evolution of $\gamma$ in ACE equality constrained networks. **Left:** VN-DGCNN on ModelNet40; layers 0, 5, 11, 13 decay to zero, with deeper layers briefly oscillating, evidence that relaxing equivariance in later layers improves the final solution. **Right:** EGNO on CMU MoCap (Run); layers 0, 2, 4, 5 hover around zero without converging, showing that strict equivariance obtained from training with equality constraints is too restrictive.

| DGCNN Variant | Class Acc. ↑ | Inst. Acc. ↑ |
|---|---|---|
| DGCNN-VN [45, 68] | $0.7707_{\pm 0.0223}$ | $0.8223_{\pm 0.0074}$ |
| DGCNN-VN$^{\text{ACE}\dagger}_{f^{\text{eq}}+f^{\text{neq}}}$ | $0.7919_{\pm 0.0156}$ | $0.8392_{\pm 0.0115}$ |
| DGCNN-VN$^{\text{ACE}\dagger}_{f^{\text{eq}}}$ | $\mathbf{0.7985}_{\pm 0.0232}$ | $\mathbf{0.8400}_{\pm 0.0123}$ |

Figure 7: Accuracy and robustness of normal and equality-constrained VN-DGCNNs. **Left:** On the clean ModelNet40, partially and purely equivariant constrained models (ACE†) outperform the baseline. **Right:** On noisy ModelNet40, training without ACE is unstable, while the equivariant model obtained with ACE ($f^{\text{eq}}$) yields a stable, high-accuracy solution.

of training (Figure 6, left). In the early layers, $\gamma$ rapidly decays to zero; in the deeper layers, it oscillates around zero, indicating that the optimizer initially explores non-equivariant directions before settling on a strictly equivariant solution that aligns with the data symmetries. In contrast, the same coefficients in EGNO oscillate around 0 throughout training on the MoCap dataset, suggesting that strict equivariance may be overly restrictive for this task (Figure 6, right). The learned slack values $u_k$ in the resilient model further support this view: as shown in Figure 5, early layers acquire larger slack, allowing flexibility where local structure dominates, while later layers remain nearly equivariant. As training progresses, the constraints tighten for all layers, increasingly penalizing non-equivariant solutions. We provide plots for all the layers in Appendix A.8. These dynamics demonstrate the ability of the proposed constrained learning framework to guide the model toward suitable, approximately equivariant behavior, stabilizing training under degradation, and improving performance over fully equivariant models.

### 5.3 Theory-Empirics Alignment

To address **RQ3**, we analyze how equivariance aligns with observed model behavior in the ACE resilient setting. We compute the equivariance deviation as $\mathbb{E}_{g \sim \text{SO}(3)} \left[ \left\| \rho_Y(g) f_{\theta,\gamma}(x) - f_{\theta,\gamma}\left(\rho_X(g)x\right) \right\| \right]$ (cf. Theorem 4.2), where the expectation is approximated using five random samples $g \in \text{SO}(3)$. As shown in Figure 4, the deviation from equivariance measured for the resilient model steadily decreases and approaches values near zero by the end of training. This suggests that the equivariance constraint is effectively enforced in practice, and that partial equivariance can be preserved in a stable and interpretable way throughout training.

## 6 Limitations and Further Work

While our ACE framework is broadly applicable to a range of architectures, it does incur some overhead: during training we introduce extra (non-equivariant) MLP layers, which modestly slow down training (see Appendix Table 4). Addressing this, one avenue for future research is to "break" equivariance without adding any auxiliary parameters, e.g., by integrating constraint enforcement directly into existing layers or by designing specialized update rules that can break equivariance while preserving runtime efficiency.

Table 2: Error metrics for equality-constrained models. **Left:** N-body, equality-constrained SEGNN projections reach the lowest test MSE ($\times 10^{-3}$). ACE† represents models trained with equality constraints, $\ast$ are trained without ACE, but with $SO(3)$ augmentations. **Right:** QM9, constraining SchNet and SEGNN reduces or matches MAE on most quantum-chemical targets.

| Method | MSE $\downarrow$ | | Target | SchNet [12] (MAE $\downarrow$) | | SEGNN [44] (MAE $\downarrow$) | |
|---|---|---|---|---|---|---|---|
| | | | | Base | ACE ($f^{eq}$) | Base | ACE ($f^{eq}$) |
| Linear [43] | 81.9 | | $\mu$ | $0.0511_{\pm 0.0011}$ | $\mathbf{0.0393}_{\pm \mathbf{0.0007}}$ | $0.0371_{\pm 0.0002}$ | $\mathbf{0.0362}_{\pm \mathbf{0.0007}}$ |
| SE(3)-Tr. [42] | 24.4 | | $\alpha$ | $0.1236_{\pm 0.0037}$ | $\mathbf{0.0952}_{\pm \mathbf{0.0042}}$ | $0.0991_{\pm 0.0033}$ | $\mathbf{0.0936}_{\pm \mathbf{0.0126}}$ |
| RF [62] | 10.4 | | $\varepsilon_{homo}$ | $0.0500_{\pm 0.0006}$ | $\mathbf{0.0443}_{\pm \mathbf{0.0006}}$ | $0.0250_{\pm 0.0005}$ | $\mathbf{0.0244}_{\pm \mathbf{0.0007}}$ |
| ClofNet [70] | 6.5 | | $\varepsilon_{lumo}$ | $0.0421_{\pm 0.0005}$ | $\mathbf{0.0360}_{\pm \mathbf{0.0004}}$ | $\mathbf{0.0244}_{\pm \mathbf{0.0003}}$ | $0.0257_{\pm 0.0006}$ |
| EGNN [43] | 7.1 | | $\Delta\varepsilon$ | $0.0758_{\pm 0.0008}$ | $\mathbf{0.0686}_{\pm \mathbf{0.0011}}$ | $\mathbf{0.0450}_{\pm \mathbf{0.0009}}$ | $0.0505_{\pm 0.0091}$ |
| EGNO [46] | 5.4 | | $\langle R^2 \rangle$ | $1.4486_{\pm 0.1225}$ | $\mathbf{0.7535}_{\pm \mathbf{0.0426}}$ | $1.2028_{\pm 0.0827}$ | $\mathbf{1.1681}_{\pm \mathbf{0.1890}}$ |
| SEGNN [44] | $5.6_{\pm 0.25}$ | | ZPVE | $0.0029_{\pm 0.0003}$ | $\mathbf{0.0018}_{\pm \mathbf{0.0000}}$ | $0.0029_{\pm 0.0004}$ | $\mathbf{0.0025}_{\pm \mathbf{0.0001}}$ |
| SEGNN$_{Rel.}$ [41] | $4.9_{\pm 0.18}$ | | $U_0$ | $9.6429_{\pm 3.2673}$ | $\mathbf{3.4039}_{\pm \mathbf{0.2872}}$ | $2.6163_{\pm 0.4020}$ | $\mathbf{2.5597}_{\pm \mathbf{0.7650}}$ |
| SEGNN$^*_{f^{eq}}$ | $5.7_{\pm 0.21}$ | | $U$ | $8.3412_{\pm 2.8914}$ | $\mathbf{3.3540}_{\pm \mathbf{0.5006}}$ | $2.5645_{\pm 0.5310}$ | $\mathbf{2.4796}_{\pm \mathbf{0.1440}}$ |
| SEGNN$^*_{f^{eq}+f^{neq}}$ | $\mathbf{3.8}_{\pm \mathbf{0.06}}$ | | $H$ | $9.5835_{\pm 4.1029}$ | $\mathbf{3.4403}_{\pm \mathbf{0.4403}}$ | $2.7577_{\pm 0.7910}$ | $\mathbf{2.4569}_{\pm \mathbf{0.5460}}$ |
| SEGNN$^{ACE\dagger}_{f^{eq}+f^{neq}}$ | $\mathbf{3.8}_{\pm \mathbf{0.16}}$ | | $G$ | $8.5405_{\pm 2.0311}$ | $\mathbf{3.6473}_{\pm \mathbf{0.4100}}$ | $2.7505_{\pm 0.6970}$ | $\mathbf{2.5954}_{\pm \mathbf{1.0250}}$ |
| SEGNN$^{ACE\dagger}_{f^{eq}}$ | $\mathbf{3.8}_{\pm \mathbf{0.17}}$ | | $c_v$ | $\mathbf{0.0379}_{\pm \mathbf{0.0003}}$ | $0.0376_{\pm 0.0006}$ | $\mathbf{0.0389}_{\pm \mathbf{0.0038}}$ | $0.0403_{\pm 0.0050}$ |

Moreover, our experiments have focused on symmetry groups acting on point clouds, molecules, and motion trajectories. A natural extension is to apply our approach to other symmetry groups and data modalities, such as 2D image convolutions or 2D graph neural networks. As an initial step in this direction, our $C_4$-symmetric 2D convolution toy experiment (see Appendix A.4) shows that ACE adapts appropriately, driving $\gamma$ toward zero when the target matches the imposed symmetry, and increases $\gamma$ when correcting for symmetry mis-specification, suggesting feasibility for broader 2D settings. Alongside this, co-designing network architectures that anticipate the dynamic tightening and loosening of equivariance constraints could produce models that converge even faster, require fewer samples, or require fewer extra parameters.

Finally, the dual variables learned by our algorithm encode where and when symmetry is most beneficial; exploiting these variables for adaptive pruning or sparsification of non-equivariant components could also lead to more efficient models.

This work aims to investigate how training can be improved for equivariant neural networks, offering benefits in areas such as drug discovery and materials science. We do not foresee any negative societal consequences.

# 7 Conclusion

We introduced *Adaptive Constrained Equivariance* (ACE), a homotopy-inspired constrained equivariance training framework, that learns to enforce equivariance layer by layer via modulation coefficients $\gamma$. Theoretically, we show how our constraints can lead to equivariant models, deriving explicit bounds on the approximation error for fully equivariant models obtained via equality constraints and on the equivariance error for partially equivariant models obtained from inequality constraints. Across N-body dynamics, molecular regression, shape recognition and motion-forecasting datasets, ACE consistently improves accuracy, sample efficiency and convergence speed, while remaining robust to input dropout and requiring no *ad hoc* schedules or domain-specific penalties. These findings suggest that symmetry can be treated as a "negotiable" inductive bias that is tightened when it helps generalization and relaxed when optimization requires flexibility, offering a practical middle ground between fully equivariant and unrestricted models. Extending the framework to other symmetry groups and modalities (such as 2D graphs and images), designing architectures that anticipate the constraint dynamics, and exploiting the dual variables for adaptive pruning of non-equivariant layers are natural next steps. Overall, we believe that allowing networks to adjust their level of symmetry during training provides a solid basis for further progress in training equivariant networks.

## 8    Acknowledgements

A. Manolache and M. Niepert acknowledge funding by Deutsche Forschungsgemeinschaft (DFG, German Research Foundation) under Germany's Excellence Strategy - EXC 2075 – 390740016, the support by the Stuttgart Center for Simulation Science (SimTech), the International Max Planck Research School for Intelligent Systems (IMPRS-IS), and the support of the German Federal Ministry of Education and Research (BMBF) as part of InnoPhase (funding code: 02NUK078). A. Manolache acknowledges funding by the EU Horizon project ELIAS (No. 101120237). The work of L.F.O. Chamon is supported by the Agence Nationale de la Recherche (ANR) project ANR-25-CE23-3477-01 as well as a chair from Hi!PARIS, funded in part by the ANR AI Cluster 2030 and ANR-22-CMAS-0002. A. Manolache is grateful to B. Luchian for insightful discussions on symmetries and equivariant functions.

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

## A Appendix / Supplementary Materials

### A.1 Pseudo-Code for ACE

To incorporate ACE, we extend a standard equivariant model with additional parameters $\gamma_i$ and non-equivariant layers. The code below illustrates how to implement the architecture accordingly:

```
class Model(nn.Module):
    def __init__(...):
        (...)
        self.gammas = nn.ParameterList([
            nn.Parameter(torch.tensor(1.)) for _ in range(n_layers)
            ])
        self.f_neq = nn.ModuleList([
            NoneqModule(...) for _ in range(n_layers)
            ])
        self.f_eq = nn.ModuleList([
            EqModule(...) for _ in range(n_layers)
            ])

    def forward(...):
        (...)
        for i in range(len(self.gammas)):
            x = self.f_eq[i](x) + gamma[i] * self.f_neq[i](x)
        (...)
```

Training requires introducing dual variables and a corresponding dual optimizer, while keeping the primal optimizer unchanged. The snippet below shows how to add these components and update them during training:

```
gammas = model.get_gammas()
dual_parameters = []
lambdas = [
    torch.zeros(1, dtype=torch.float, requires_grad=False).squeeze()
    for _ in gammas
    ]
dual_parameters.extend(lambdas)

# If we want to use Algorithm 2, add:
# us = [
    nn.Parameter(torch.zeros(1, requires_grad=True).squeeze())
    for _ in range(len(lambdas))
    ]
# dual_parameters.extend(us)

optimizer_primal = optim.Adam/SGD/etc(model.parameters())
optimizer_dual = optim.Adam/SGD/etc(dual_parameters)

# Now, in the training routine:
(...)
loss = loss_fn(preds, y)
dual_loss = 0
slacks = [gamma for gamma in gammas]

for dual_var, slack in zip(lambdas, slacks):
    dual_loss += dual_var * slack

# Explicit, but can also be obtained via autodiff
for i, slack in enumerate(slacks):
    lambdas[i].grad = -slack

# If we use Algorithm 2, change to:
# loss = loss + 1/2 * torch.norm(torch.stack(us)) ** 2
# dual_loss = 0
```

```
# slacks = [torch.norm(gamma) - u for gamma, u in zip(gammas, us)]
# for dual_var, slack, u in zip(lambdas, slacks, us):
# dual_loss += (dual_var * slack) - (u * slack)
# for i, (slack, u) in enumerate(zip(slacks, us)):
# lambdas[i].grad = -slack - u

lagrangian = loss + dual_loss
lagrangian.backward()
optimizer.step()
optimizer_dual.step()

# If we use Algorithm 2, add:
# for lambda in lambdas:
# lambda.clamp_(min=0)
```

(a) Test MSE ($\times 10^{-3}$) on the N-Body dataset under different initializations of $\gamma$. "N/A" indicates that ACE is not used.

| $\eta_d$ | $\gamma_{\text{init}}$ | Test MSE |
|---|---|---|
| N/A | N/A | $6.64 \pm 0.14$ |
| $8 \times 10^{-4}$ | 1 | $4.97 \pm 0.01$ |
| $8 \times 10^{-4}$ | 0 | $5.39 \pm 0.33$ |

(b) Test MSE ($\times 10^{-3}$) on the N-Body dataset for different dual learning rates $\eta_d$. "N/A" indicates that ACE is not used. The primal learning rate is fixed to $\eta_p = 9 \times 10^{-4}$.

| $\eta_d$ | Test MSE |
|---|---|
| N/A | $6.64 \pm 0.14$ |
| $1 \times 10^{-5}$ | $6.10 \pm 0.25$ |
| $8 \times 10^{-5}$ | $5.47 \pm 0.29$ |
| $1 \times 10^{-4}$ | $5.24 \pm 0.29$ |
| $8 \times 10^{-4}$ | $4.97 \pm 0.01$ |
| $1 \times 10^{-3}$ | $5.03 \pm 0.09$ |
| $8 \times 10^{-3}$ | $5.11 \pm 0.16$ |

Table 3: Ablations on the N-Body dataset. (a) Effect of initializing $\gamma$. (b) Effect of varying the dual learning rate $\eta_d$.

## A.2 Discussion regarding gamma initialization

All experiments in the main text use $\gamma_{\text{init}} = 1$, as we did not find it necessary to run a separate hyperparameter search over this value. To illustrate the effect of initialization, we performed an ablation with SEGNN on the N-Body dataset where we compared $\gamma_{\text{init}} = 0$ against $\gamma_{\text{init}} = 1$ (Table 3a).

As can be seen, both models using ACE obtain improved performance compared to removing ACE altogether ($\eta_d = $ N/A, $\gamma_{\text{init}} = $ N/A). However, initializing with $\gamma_{\text{init}} = 1$ leads to the lowest test error. We observe empirically that that zero initialization encourages faster convergence in the early stages of training, but could bias the optimization toward solutions that closely adhere to the symmetry constraints, potentially limiting exploration. In contrast, initializing $\gamma_{\text{init}} = 1$ gives us slower initial convergence but ultimately achieves better final accuracy.

## A.3 Relationship between primal and dual learning rates

The dual optimization in ACE introduces a learning rate $\eta_d$ in addition to the primal learning rate $\eta_p$. We observe empirically that setting $\eta_d$ close to $\eta_p$ leads to stable convergence across architectures and datasets. From a practical standpoint, it is often sufficient to first tune $\eta_p$ and then either set $\eta_d = \eta_p$ or choose a slightly smaller value (e.g., $\eta_d = 8 \times 10^{-4}$ when $\eta_p = 9 \times 10^{-4}$). While default choices for $\eta_p$ (as in official implementations) tend to perform well, we find that $\eta_d$ is less sensitive and can be specified by simple proportional scaling. To quantify this effect, we conducted a sensitivity study with SEGNN on the N-Body dataset with 1000 samples. Table 3b reports the test MSE (mean $\pm$ std) under different $\eta_d$ values, including a baseline without ACE ("N/A"). The best performance was consistently achieved when $\eta_d \approx \eta_p$, confirming that matching the dual and primal learning rates provides a robust default setting.

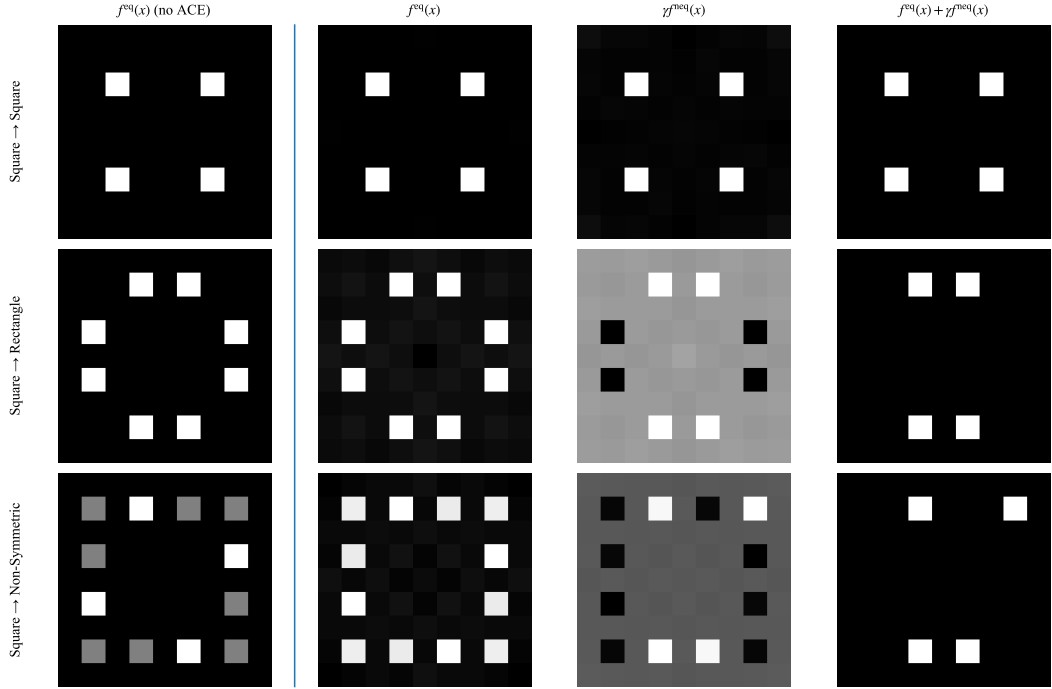

Figure 8: $C_4$ symmetry toy experiment with ACE (resilient constraints). We want to transform a square to other shapes. The strictly equivariant baseline succeeds only when the target shares $C_4$ symmetry (top row). Under ACE with resilient constraints, the learned coefficient $\gamma$ adapts: in the aligned case, $\gamma$ decreases towards zero, and the outputs match; in the mismatched cases (second and third rows), $\gamma$ increases, adding a missing signal that corrects the symmetry mis-specification. For visualization, outputs are min–max normalized per panel.

## A.4 Correcting symmetry mis-specification with ACE

Motivated by analyses of symmetry breaking in 2D CNNs [27], we take a different route — we do not use relaxed group convolutions; instead, we keep the convolution strictly $C_4$–equivariant and relax equivariance via ACE by adding a non-equivariant branch $\gamma f^{\mathrm{neq}}$ to an equivariant $C_4$ CNN $f^{\mathrm{eq}}$.

We study three image-to-image mappings that differ in how they align with the imposed $C_4$ symmetry: (i) **Square→Square** (aligned target; a $C_4$-symmetric pattern), (ii) **Square→Rectangle** (mismatched symmetry; target requires $C_2$), and (iii) **Square→Non-Symmetric** (no global symmetry). We train a strictly equivariant baseline ($f^{\mathrm{eq}}$, "no ACE") and an ACE model $f^{\mathrm{eq}} + \gamma f^{\mathrm{neq}}$ under resilient (inequality) constraints. In Figure 8, we see an ACE model output decomposed into $f^{\mathrm{eq}}(x)$, the correction $\gamma f^{\mathrm{neq}}(x)$, and their sum. The baseline solves Square→Square but fails for the mis-specified cases. With ACE, the learned coefficient behaves as desired: in Square→Square, $\gamma$ decays towards 0 and the outputs of the models are similar, indicating that the imposed symmetry matches the task; in Square→Rectangle and Square→Non-Symmetric, $\gamma$ grows, obtaining a non-equivariant correction that recovers the desired outputs. A natural direction for future work is to evaluate ACE across modern 2D equivariant architectures on real datasets and quantifying accuracy-equivariance trade-offs.

## A.5 Controlling the scale of the non-equivariant component.

The non-equivariant layer $f_\theta^{\mathrm{neq},i}$ can in principle dominate the constraint $\gamma_i$ if its output norm grows too large. To prevent this, we assume that each such layer is $M$-Lipschitz and a bounded operator, i.e., $\|f_\theta^{\mathrm{neq},i}(x)\| \le B\|x\|$ for all inputs $x$ during training. This condition can be enforced by reparameterizing the weights, for example by using Spectral Normalization [71], which guarantees $B \le 1$.

From a practical standpoint, we found that enforcing exact equivariance (Algorithm 1) did not require explicit use of Spectral Normalization: during model selection, we checkpoint based on the validation performance of the equivariant projection $f_\theta^{\text{eq}}$, which naturally disregards the non-equivariant part. However, in the case of Resilient Constrained Learning (Algorithm 2), Spectral Normalization becomes important for reducing the equivariance error of $f_\theta^{\text{eq}} + f_\theta^{\text{neq}}$ (see Fig. 4). This is a consequence of $B$ not being bounded if the weights of $f_\theta^{\text{neq},i}$ are not reparameterized. Hence, Spectral Normalization provides a simple and effective mechanism to control the scale of the non-equivariant branch and to reduce the equivariance error.

### A.6 Proof of Theorem 4.1

Here, we prove the more refined bound below:

**Theorem A.1.** *Consider $f_{\theta,\gamma}$ as in (3) satisfying Assumption 1. Then,*

$$\left\| f_{\theta,\gamma}(x) - f_{\theta,0}(x) \right\| \leq \left[ \sum_{k=0}^{L-1} |\gamma_{k+1}| \left( 1 + \frac{1}{k} \sum_{j=1}^{k} |\gamma_j| \right)^k \right] B M^{L-1} \|x\|, \quad \text{for all } x \in X.$$

The bound in Theorem 4.1 can be obtained directly from the above using the fact that $\gamma_k \leq \bar{\gamma}$.

*Proof.* The proof proceeds by bounding the error layer by layer. Indeed, begin by defining

$$z_0 = x \quad \text{and} \quad z_i = f_{\theta,\gamma}^i(z_{i-1}) = f_\theta^{\text{eq},i}(z_{i-1}) + \gamma_i f_\theta^{\text{neq},i}(z_{i-1}), \text{ for } i = 1, \ldots, L, \tag{4}$$

so that $z_L = f_{\theta,\gamma}(x)$. Likewise, let

$$z_0^{\text{eq}} = x \quad \text{and} \quad z_i^{\text{eq}} = f_\theta^{\text{eq},i}(z_{i-1}^{\text{eq}}), \text{ for } i = 1, \ldots, L, \tag{5}$$

so that $z_L^{\text{eq}} = f_{\theta,0}(x)$.

Proceeding, we use the triangle equality to obtain

$$\delta_i \triangleq \|z_i - z_i^{\text{eq}}\| \leq \left\| f_i^{\text{eq}}(z_{i-1}) - f_i^{\text{eq}}(z_{i-1}^{\text{eq}}) \right\| + |\gamma_i| \left\| f_i^{\text{neq}}(z_{i-1}) \right\| \tag{6}$$

We bound (6) using the fact that $f_i^{\text{eq}}$ is $M$-Lipschitz continuous and that $f_i^{\text{neq}}$ is a bounded operator (Ass. 1) to write

$$\left\| f_i^{\text{eq}}(z_{i-1}) - f_i^{\text{eq}}(z_{i-1}^{\text{eq}}) \right\| \leq M \left\| z_{i-1} - z_{i-1}^{\text{eq}} \right\| = M\delta_{i-1}$$
$$|\gamma_i| \left\| f_i^{\text{neq}}(z_{i-1}) \right\| \leq |\gamma_i| B \| z_{i-1} \|.$$

Back in (6) we get

$$\delta_i \leq M\delta_{i-1} + |\gamma_i| B \| z_{i-1} \|. \tag{7}$$

Noticing that the quantity of interest is in fact $\left\| f_{\theta,\gamma}(x) - f_{\theta,0}(x) \right\| = \|z_L - z_L^{\text{eq}}\| = \delta_L$, we can use (7) recursively to obtain

$$\delta_L \leq B \sum_{k=1}^{L} M^{L-k} |\gamma_k| \| z_{k-1} \|, \tag{8}$$

using the fact that $\delta_0 = 0$.

To proceed, note from Assumption 1 and the definition of the architecture in (3) that $f_{\theta,\gamma}^i$ is $M(1 + |\gamma_i|)$-Lipschitz and $z_i = f_{\theta,\gamma}^i \circ \cdots \circ f_{\theta,\gamma}^1$. Hence, we obtain

$$\| z_i \| \leq \prod_{j=1}^{i} \left[ M(1 + |\gamma_j|) \right] \|x\|, \quad \text{for } i = 1, \ldots, L. \tag{9}$$

Using the fact that $z_0 = x$, (8) immediately becomes

$$\delta_L \leq B M^{L-1} \left[ |\gamma_1| + \left( \sum_{k=2}^{L} |\gamma_k| \prod_{j=1}^{k-1} (1 + |\gamma_j|) \right) \right] \|x\|, \tag{10}$$

To obtain the bound in Theorem A.1, suffices it to use the relation between the arithmetic and the geometric mean (AM-GM inequality) to obtain

$$\prod_{j=1}^{k-1}(1+|\gamma_j|) \leq \left(1 + \frac{1}{k-1}\sum_{\ell=1}^{k-1}|\gamma_\ell|\right)^{k-1}.$$

Rearranging the terms yields the desired result. $\qquad\square$

## A.7 Proof of Theorem 4.2

Once again we prove a more refined version of Theorem 4.2:

**Theorem A.2.** *Consider $f_{\theta,\gamma}$ as in* (3) *satisfying Assumption 1 and let $C = \max(B/M, 1)$*

$$\left\|\rho_Y(g)f_{\theta,\gamma}(x) - f_{\theta,\gamma}\big(\rho_X(g)x\big)\right\| \leq 2\sum_{k=1}^{L}|\gamma_k|\left(1 + \frac{C}{L-1}\sum_{j\neq k}|\gamma_j|\right)^{L-1}B^2 M^{L-1}\|x\|,$$

*for all $x \in X$ and $g \in G$.*

The result in Theorem 4.2 is recovered by using the fact that $\gamma_i \leq \bar{\gamma}$.

*Proof.* We proceed once again in a layer-wise fashion using the definitions in (4) and

$$\tilde{z}_0 = \rho_X(g)x \quad \text{and} \quad \tilde{z}_i = f_{\theta,\gamma}^i(\tilde{z}_{i-1}) = f_\theta^{\text{eq},i}(\tilde{z}_{i-1}) + \gamma_i f_\theta^{\text{neq},i}(\tilde{z}_{i-1}), \text{ for } i = 1,\dots,L.$$

Hence, we can write the equivariance error at the $i$-th layer as

$$\epsilon_i \triangleq \|\rho_i(g)z_i - \tilde{z}_i\| = \left\|\rho_i(g)f_\theta^{\text{eq},i}(z_{i-1}) + \gamma_i\rho_i(g)f_\theta^{\text{neq},i}(z_{i-1}) - f_\theta^{\text{eq},i}(\tilde{z}_{i-1}) - \gamma_i f_\theta^{\text{neq},i}(\tilde{z}_{i-1})\right\|$$

Notice that, in this notation, we seek a bound on $\epsilon_L$ independent of $x \in X$ and $G$.

Proceeding layer-by-layer, use the equivariance of $f^{\text{eq},i}$ and the triangle inequality to bound $\epsilon_i$ by

$$\epsilon_i \leq \left\|f_\theta^{\text{eq},i}\big(\rho_{i-1}(g)z_{i-1}\big) - f_\theta^{\text{eq},i}(\tilde{z}_{i-1})\right\| + |\gamma_i|\left\|\rho_i(g)f_\theta^{\text{neq},i}(z_{i-1}) - f_\theta^{\text{neq},i}(\tilde{z}_{i-1})\right\| \quad (11)$$

Using the fact that $f^{\text{eq},i}$ is Lipschitz continuous and $f^{\text{neq},i}$ is a bounded operator (Assumption 1), we obtain

$$\left\|f_\theta^{\text{eq},i}\big(\rho_{i-1}(g)z_{i-1}\big) - f_\theta^{\text{eq},i}(\tilde{z}_{i-1})\right\| \leq M\|\rho_{i-1}(g)z_{i-1} - \tilde{z}_{i-1}\| = M\epsilon_{i-1}$$

$$\left\|\rho_i(g)f_\theta^{\text{neq},i}(z_{i-1}) - f_\theta^{\text{neq},i}(\tilde{z}_{i-1})\right\| \leq B(B\|z_{i-1}\| + \|\tilde{z}_{i-1}\|)$$

We can further expand the second bound using the fact that

$$\|\tilde{z}_{i-1}\| = \|\tilde{z}_{i-1} - \rho_{i-1}z_{i-1} + \rho_{i-1}z_{i-1}\| \leq \|\tilde{z}_{i-1} - \rho_{i-1}z_{i-1}\| + \|\rho_{i-1}z_{i-1}\| \leq \epsilon_{i-1} + B\|z_{i-1}\|.$$

When combined into (11) we obtain the bound

$$\epsilon_i \leq (M + |\gamma_i|B)\epsilon_{i-1} + 2|\gamma_i|B^2\|z_{i-1}\|. \quad (12)$$

Applying (12) recursively, we obtain

$$\epsilon_L \leq 2B^2\sum_{k=1}^{L}|\gamma_k|\left[\prod_{j=k+1}^{L}(M + |\gamma_j|B)\right]\|z_{k-1}\|, \quad (13)$$

where we used the fact that $\epsilon_0 = \|\rho_X(g)z - \tilde{z}_0\| = 0$.

To proceed, we once again use Assumption 1 and the definition of the architecture in (3) to obtain (9), so that (13) becomes

$$\epsilon_L \leq 2B^2\sum_{k=1}^{L}|\gamma_k|\left[\prod_{j=k+1}^{L}(M + |\gamma_j|B)\right]\left[\prod_{j=1}^{k-1}\big[M(1 + |\gamma_j|)\big]\right]\|x\|.$$

Table 4: Parameter counts and runtimes for the equivariant baselines versus models trained with ACE. We relax equivariance during training with extra linear or MLP layers, operations that PyTorch executes efficiently, therefore training and validation are only marginally slower than the baselines. At inference we drop these non-equivariant layers, yielding virtually identical runtimes to the pure equivariant models. All experiments were performed on a RTX A5000 GPU with an Intel i9-11900K CPU.

| Model | #Parameters (Train) | Train time (s/epoch) | Val time (s/epoch) |
|---|---|---|---|
| SEGNN - N-body (vanilla) | 147,424 | $1.368_{\pm 0.004}$ | $0.505_{\pm 0.005}$ |
| SEGNN - N-body (constrained) | 937,960 | $1.512_{\pm 0.004}$ | $0.527_{\pm 0.004}$ |
| SEGNN - QM9 (vanilla) | 1,033,525 | $118.403_{\pm 0.108}$ | $11.152_{\pm 0.083}$ |
| SEGNN - QM9 (constrained) | 9,869,667 | $129.039_{\pm 0.043}$ | $11.402_{\pm 0.089}$ |
| EGNO - MoCap (vanilla) | 1,197,176 | $0.702_{\pm 0.007}$ | $0.130_{\pm 0.000}$ |
| EGNO - MoCap (constrained) | 1,304,762 | $1.106_{\pm 0.004}$ | $0.130_{\pm 0.000}$ |
| DGCNN-VN - ModelNet40 (vanilla) | 2,899,830 | $57.826_{\pm 0.500}$ | $5.954_{\pm 0.043}$ |
| DGCNN-VN - ModelNet40 (constrained) | 8,831,612 | $81.760_{\pm 0.169}$ | $5.978_{\pm 0.006}$ |
| SchNet - QM9 (vanilla) | 127,553 | $10.389_{\pm 0.152}$ | $0.944_{\pm 0.003}$ |
| SchNet - QM9 (constrained) | 153,543 | $11.126_{\pm 0.155}$ | $0.947_{\pm 0.002}$ |

To simplify the bound, we take $C = \max(B/M, 1)$, so that we can write

$$
\begin{aligned}
\epsilon_L &\leq 2M^{L-1}B^2 \sum_{k=1}^{L} |\gamma_k| \left[ \prod_{j=k+1}^{L} \left( 1 + \frac{B}{M}|\gamma_j| \right) \right] \left[ \prod_{j=1}^{k-1} (1 + |\gamma_j|) \right] \|x\| \\
&\leq 2M^{L-1}B^2 \sum_{k=1}^{L} |\gamma_k| \left[ \prod_{j\neq k} (1 + C|\gamma_j|) \right] \|x\|,
\end{aligned}
\tag{14}
$$

Using once again the relation between the arithmetic and the geometric mean (AM-GM inequality) yields

$$
\epsilon_L \leq 2 \sum_{k=1}^{L} |\gamma_k| \left( 1 + \frac{C}{L-1} \sum_{j\neq k} |\gamma_j| \right)^{L-1} B^2 M^{L-1} \|x\|.
$$

$\square$

## A.8 Additional plots for $\gamma$ and $\lambda$

While the main text characterizes only the evolution of the modulation coefficients $\gamma_i$, it omits the corresponding dual variables $\lambda_i$, which serve as the Lagrange multipliers enforcing each equivariance constraint. Here, we present the full, detailed plots of $\gamma_i$ and $\lambda_i$ for ModelNet40 (Fig. 9 and Fig. 10) and CMU MoCap (Fig. 11 and Fig. 12), allowing simultaneous inspection of the symmetry relaxation and the accompanying optimization "pressure". By tracing $\lambda_i$ alongside $\gamma_i$, one can observe how retained constraint violations accumulate multiplier weight, driving subsequent adjustments to $\gamma_i$ and thereby revealing the continuous "negotiation" between inductive bias and training flexibility instantiated by our primal–dual algorithm.

## A.9 Additional Experimental Details

All experiments are implemented in PyTorch [72] using each model's official code base and default hyperparameters as a starting point. Because several repositories employed learning rates that proved sub-optimal, we performed a grid search for both the primal optimizer and, when applicable, the dual optimizer in our constrained formulation. After selecting the best values, we re-ran every configuration with multiple random seeds. A noteworthy mention is that the test results on the QM9 dataset reported in Table 2 use different splits for the training, validation and test datasets. For SchNet, we shuffle the dataset obtained from PyTorch Geometric [73] and split the dataset with a 80%/10%/10% train/validation/test ratio. For SEGNN, we use the official splits from [44].

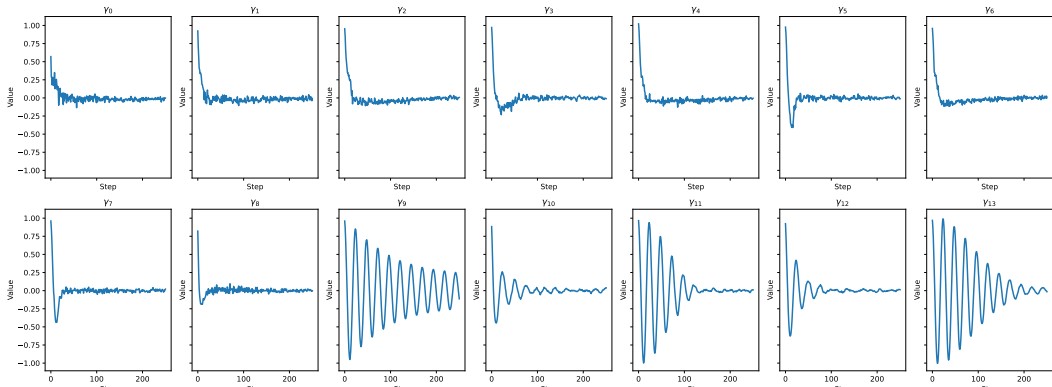

Figure 9: Full layer-wise $\gamma_i$ values on ModelNet40. The ACE VN-DGCNN quickly drives the modulation coefficients $\gamma_i$ of the first few layers towards 0, locking those layers into fully equivariant behaviour, while deeper layers exhibit brief oscillations before convergence.

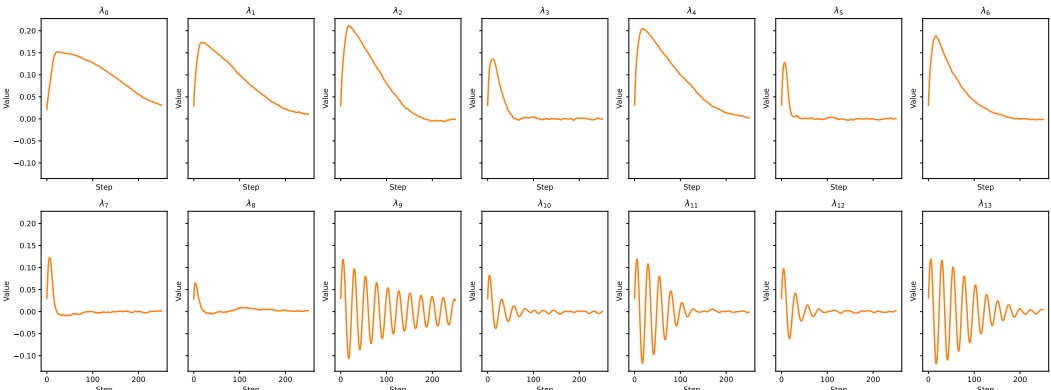

Figure 10: Dual variables $\lambda_i$ on ModelNet40. The companion values of $\lambda_i$ mirror the $\gamma_i$ dynamics: layers whose $\gamma_i$ are further away from zero accumulate larger $\lambda_i$, signaling higher constraint cost.

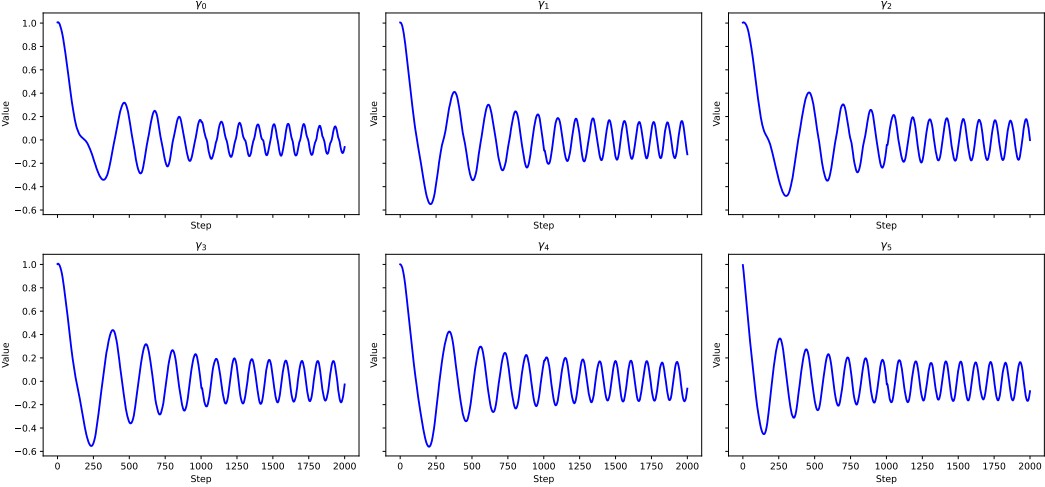

Figure 11: Layer-wise $\gamma_i$ dynamics on CMU MoCap (Subject #9, Run) dataset. Rather than decaying to zero as on the ModelNet40 dataset, the modulation coefficients $\gamma_i$ oscillate around zero for the full 2000-step training window, signaling that imposing strict equivariance is overly restrictive for the dataset and model.

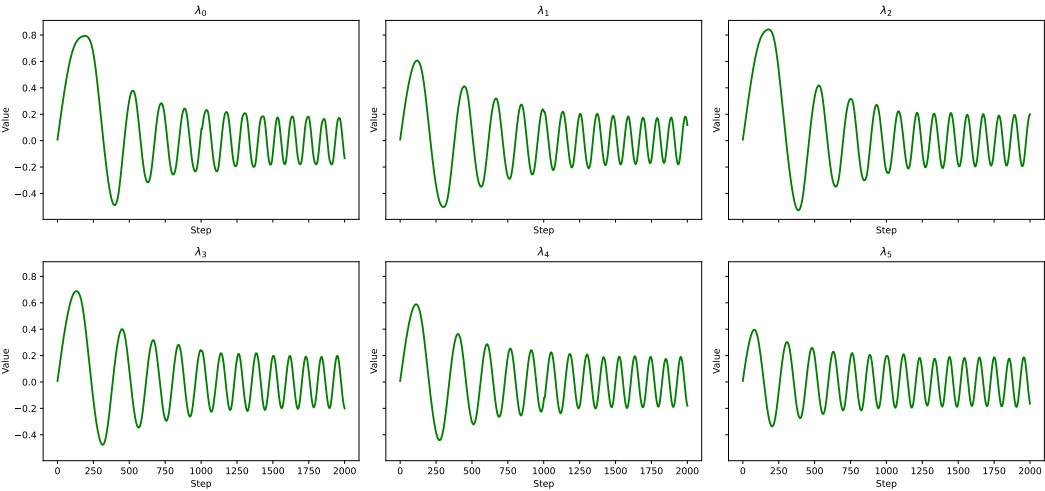

Figure 12: The dual variables $\lambda_i$ on CMU MoCap (Subject #9, Run). Similar to the $\gamma_i$ coefficients, the Lagrange multipliers $\lambda_i$ oscillate near zero, but never converge.

In the case of VN-DGCNN, we followed the setup of Pertigkiozoglou et al. [41], which trains equivariant models on a sub-sampled resolution of 300 points per point cloud. This differs from the original VN-DGCNN paper [45], where 1024 points are used. Moreover, our implementation only tuned the primal and dual learning rates ($\eta_p, \eta_d$), without modifying other components or performing a broader hyperparameter search. Due to these differences, both in input resolution and in the extent of tuning, the reported results are not directly comparable to those in Deng et al. [45].

Table 5 lists the learning rates explored and the number of seeds per dataset.

The code and instructions on how to reproduce the experiments are publicly available at `https://github.com/andreimano/ACE`.

Table 5: Learning rates and number of random seeds used for each dataset.

| Dataset | Primal LR grid | Dual LR grid | Seeds |
|---|---|---|---|
| ModelNet40 | $\{\,0.10,\ 0.20\,\}$ | $\{\,1{\times}10^{-4},\ 1{\times}10^{-3}\,\}$ | 5 |
| CMU MoCap | $\{\,5{\times}10^{-4},\ 1{\times}10^{-3}\,\}$ | $\{\,5{\times}10^{-5},\ 2{\times}10^{-4},\ 1{\times}10^{-2}\,\}$ | 3 |
| N-Body | $\{\,5{\times}10^{-4},\ 8{\times}10^{-4},\ 9{\times}10^{-4},\ 1{\times}10^{-3},\ 5{\times}10^{-3}\,\}$ | $\{\,5{\times}10^{-5},\ 1{\times}10^{-4},\ 8{\times}10^{-4}\,\}$ | 5 |
| QM9 | $\{\,5{\times}10^{-5},\ 5{\times}10^{-4},\ 5{\times}10^{-3},\ 1{\times}10^{-2}\,\}$ | $\{\,5{\times}10^{-4},\ 1{\times}10^{-2},\ 1{\times}10^{-1}\,\}$ | 5 |

All of the experiments were performed on internal clusters that contain a mix of Nvidia RTX A5000, RTX 4090, or A100 GPUs and Intel i9-11900K, AMD EPYC 7742, and AMD EPYC 7302 CPUs. All experiments consumed a total of approximately 1000 GPU hours, with the longest compute being consumed on the QM9 dataset. Runtimes and parameter counts are reported in Table 4.

