# OpenReview forum: "Learning (Approximately) Equivariant Networks via Constrained Optimization"
_NeurIPS.cc/2025/Conference — NeurIPS 2025 oral_

### Official Review · Reviewer_bUHD · 2025-06-22

**Clarity:** 4
**Significance:** 4
**Originality:** 3
**Rating:** 5
**Confidence:** 4

**Summary:**

The paper introduces Adaptive Constrained Equivariance (ACE), a constrained optimization framework that applies layer-wise homotopy for equivariance training. The key novelty is leveraging the dual problem and tighten the constraints via dual descent. The paper proposed algorithms for strictly and partially equivariant data, and provide theoretical analysis on the optimality bound. A range of experiments demonstrate the superior performance of the proposed framework, as well as analysis on training dynamics and comparison with theoretical bounds.

**Questions:**

1. The ACE framework introduces new step size parameters $\eta_p$, $\eta_d$. How sensitive is the method to these parameters, and what are some general rules for setting these values in practice?
2. Assumption 1 is fairly strong, for example ReLU networks are non-differentiable. So in real-data scenarios where the assumptions are violated, which aspects of the framework will be most vulnerable to failure? How robust is the framework to such violations?

**Ethical Concerns:**

["NO or VERY MINOR ethics concerns only"]

**Final Justification:**

I appreciate the authors' response, and the explanation on choices of the $\eta$ parameters. Such rule of thumb results can be helpful, while in practice these parameters may still need to be tuned. Overall I will maintain my score.

**Limitations:**

Yes

**Quality:**

4

**Strengths And Weaknesses:**

Strength
1. The paper studies the well-motivated problem of equivariant network training, and provides novel insight with the proposed dual problem formulation.
2. The theorems provide concrete tight error bounds on equivariance performance.
3. Experimental evidence is rich and covers performance evaluation, model analysis and alignment with theoretical results.
4. The proposed framework is applicable to general network architectures.
5. Organization and presentation of the paper is clear.

Weakness
1. Assumption 1 is strong in practice, and having some discussions on its connection with real scenarios will be helpful.
2. The newly introduced step size parameters $\eta$ still needs tuning.

---

> ### Author Rebuttal · Authors · 2025-07-30
>
> We would like to thank Reviewer bUHD for their comments and for the positive assessment of our work. In the following, we address the Reviewer’s concerns:
>
> **W1 - Learning rate $\eta_d$ for the dual variables**: Both Reviewer bUHD and Reviewer bnrd point out that, while our method indeed doesn't require hand-crafted schedules for $\gamma$ or tuning of regularization hyperparameters, the dual problem optimization involves a new learning rate $\eta_d$. Empirically, we have observed that having a dual learning rate $\eta_d \approx \eta_p$ works well in most situations. Our recommendation from a practical perspective is to find a good learning rate $\eta_p$ and then set $\eta_d = \eta_p$, or to set $\eta_d$ to a slightly lower value (e.g. if $\eta_p = 9e^{-4}$ then set $\eta_d=8e^{-4}$). We usually had a harder time searching for $\eta_p$ than $\eta_d$, as can be seen in Supplementary Table 4. Using the default learning rates for $\eta_p$ (e.g., from the official implementation of a model) will generally produce good results, but we observed that in some cases they were suboptimal, as stated on Line 951 in the Supplementary.
>
> We have run a sensitivity analysis on the N-Body dataset (1000 samples). Please note that the table contains a baseline experiment ($\eta_d$=N/A means ACE is not used). The initial values for $\eta_p, \eta_d$ were obtained via the initial learning rate grid search (Table 4 of the Supplementary). We set $\eta_p=9e^{-4}$ for all experiments and only evaluate with the strictly equivariant projection $f_{eq}$.
>
> | $\eta_d$  | Test MSE ($\times 10^{-3}$) |
> | --------- | --------------------------- |
> | N/A       | $6.64\pm0.14$               |
> | $1e^{-5}$ | $6.10\pm0.25$               |
> | $8e^{-5}$ | $5.47\pm0.29$               |
> | $1e^{-4}$ | $5.24\pm0.29$               |
> | $8e^{-4}$ | $4.97\pm0.01$               |
> | $1e^{-3}$ | $5.03\pm0.09$               |
> | $8e^{-3}$ | $5.11\pm0.16$               |
>
> As can be seen, different learning rates $\eta_d$ can affect final performance. However, in all experiments, using ACE produced better results than the base model (N/A), with the best results being obtained when $\eta_p \approx \eta_d$.
>
> We agree with the reviewer that this is an important point. We believe that this discussion adds further insight to our method and is very useful from a practitioner's perspective. We will include the discussion together with additional plots in the next iteration of our paper.
>
> ---
>
> **W2 - Assumption 1 is strong:** The main reason why we are making Assumption 1 is that the magnitude of the output of the non-equivariant layer $f_\theta^{neq,i}$ might otherwise dominate $\gamma_i$.
>
> We would argue that the assumption is not strong, and can indeed be easily satisfied in practice - we assume that the layers are M-Lipschitz and bounded operators with $||f_\theta^{neq,i}||\leq B||x||$. This can be easily enforced by reparametrizing the weights, e.g., by using Spectral Normalization [1] (L180-181) so that $B\leq 1$, even when using ReLU non-linearities. Moreover, we **do** use ReLU non-linearities in practice when $f_\theta^{neq}$ is a MLP.
>
> From a practical standpoint, when seeking to obtain exact equivariance (Algorithm 1), we did not need to use Spectral Normalization --- when doing model selection, we checkpoint based on the validation performance of the equivariant projection $f_\theta^{eq}$, ignoring the non-equivariant component. However, we have observed that in the case of Resilient Constrained Learning (Algorithm 2), Spectral Normalization is needed for the equivariance error of $f_\theta^{eq} + f_\theta^{neq}$ to decrease (Figure 4); this is indeed due to B not being bounded if the weights are not reparametrized.
>
> We will add a more explicit discussion regarding the use of Spectral Normalization in the final version of our paper.
>
> ---
>
> Once again, we would like to kindly thank the Reviewer for taking their time to review our paper and provide constructive feedback. We believe that these points have allowed us to improve both the practical aspects and the theoretical clarity of our method. We are happy to discuss any further questions or concerns in the next stage.
>
> ---
>
> [1]: Miyato et al, "Spectral Normalization for Generative Adversarial Networks", ICLR 2018

---

> > ### Comment · Reviewer_bUHD · 2025-08-08
> >
> > I appreciate the authors' response, and the explanation on choices of the parameters and Assumption 1. Adding discussions regarding these points will strengthen the paper. In particular having some rule-of-thumb criteria backed by experiments for choosing the $\eta$ parameters is critical for the actual applicability of the proposed algorithm. Overall I will maintain my positive score.

---

### Official Review · Reviewer_k8VA · 2025-06-30

**Clarity:** 2
**Significance:** 3
**Originality:** 3
**Rating:** 5
**Confidence:** 3

**Summary:**

This paper introduces Adaptive Constrained Equivariance (ACE), a framework for handling approximately equivariant problems. Each layer is decomposed into an equivariant block plus a non-equivariant branch scaled by $\gamma$. Training starts with $\gamma=1$ (fully flexible) and gradually shrinking $\gamma$ through the constrained optimization procedure, smoothly interpolating from non-equivariant to strictly equivariant behavior. Experiments on N-body dynamics, molecular regression, shape recognition, and motion-forecasting tasks demonstrate the effectiveness of ACE.

**Questions:**

1. Would the authors consider adding a 2-D convolutional experiment that directly compares ACE with [26, 27]?
2. The current experimental section feels crowded. Splitting it into clear subsections (for example, by data domain) would improve readability.
2. For each task, could the authors explicitly state whether the ground-truth function is strictly or only approximately equivariant? Highlighting this distinction would help readers interpret the results.
4. Do all experiments initialize $\gamma$ with 1? Have the authors considered different initial $\gamma$ values, such as starting with $\gamma=0$?

**Ethical Concerns:**

["NO or VERY MINOR ethics concerns only"]

**Final Justification:**

The authors rebuttal resolved most of my questions, and it is great to hear that they are working on the 2D conv experiment. So I keep my initial score of accept.

**Limitations:**

Although the authors discuss limitations, the section is relegated to the appendix. Moving it into the main text in the final version would improve transparency and balance.

**Quality:**

3

**Strengths And Weaknesses:**

Strength
1. The methodology is generic and can be plugged into diverse backbone architectures to tackle approximately equivariant settings.
2. The approach is thoroughly evaluated across multiple domains and base architectures, showcasing broad applicability.

Weakness
1. As the authors note, there are no experiments on 2-D convolutional networks. Given the wide usage of (equivariant) CNNs, such a study would greatly strengthen the paper.
2. The paper compares against only one relaxed-equivariance baseline. Including additional methods such as [26, 27, 50] would make the empirical comparison more convincing.

---

> ### Author Rebuttal · Authors · 2025-07-30
>
> We would like to thank Reviewer k8VA for their thoughtful and constructive feedback. In the following, we address each of their concerns in detail.
>
> **Q1 - ACE with 2D convolutions:**
>
> We thank the Reviewer for this suggestion. As also noted by Reviewer hCpi, incorporating experiments with 2D equivariant convolutions would be a valuable addition to our paper.
>
> We are currently preparing experiments inspired by the Relaxed Group Convolutions introduced in Wang et al., 2024 [27], where the authors visualize the weights of a relaxed $C_4$-equivariant convolution under different symmetry groups, such as $C_2$. In a similar spirit, we plan to train a $C_4$-equivariant network using ACE and analyze how the non-equivariant component $f_\theta^{neq}$ changes across different symmetry settings.
>
> Furthermore, since $f_\theta^{neq}$ can be interpreted as a correction term to $f_\theta^{eq}$ in the presence of symmetry mismatch, we also plan to visualize and compare the contributions of $f_\theta^{eq}(x)$ and $\gamma f_\theta^{neq}(x)$ to the final prediction. These visualizations will help illustrate how ACE adapts when exact symmetry is not present.
>
> ---
>
> **Q2 - Experimental Section is too crowded:**
>
> The Reviewer has a good point. The final version of the paper will include one additional content page that we will use  to refactor the experimental section for clarity and structure. Specifically, we will reorganize the section into subsections, expand on the descriptions of each experimental setup, and provide more detailed analysis of the results.
>
> ---
>
> **Q3 - Obtaining Partially vs. Exactly Equivariant Functions; Exact Symmetries Across Experiments:**
>
> We appreciate the Reviewer’s observation and will revise the paper to clarify the distinction between exact and partial equivariance. In particular, we will specify for each experiment which symmetry assumptions are enforced exactly and which are only approximately satisfied.
>
> ---
>
> **Q4 - Initializing $\gamma$ at Different Values:**
>
> All experiments use $\gamma_{init}=1$ (we did not see the need to run any hyperparameter search for $\gamma$). Below, we show the results of an experiment on the N-Body dataset that shows the effect of using $\gamma_{init}=0$:
>
> | $\eta_d$  | $\gamma_{init}$ | Test MSE ($\times 10^{-3}$) |
> | --------- | --------------- | --------------------------- |
> | N/A       | N/A             | $6.64\pm0.14$               |
> | $8e^{-4}$ | 1               | $4.97\pm0.01$               |
> | $8e^{-4}$ | 0               | $5.39\pm0.33$               |
>
> As can be seen, both cases yield better results than not using ACE ($\eta_d=N/A, \gamma_{init}=N/A$). However, better performance is obtained when starting in an unconstrained scenario ($\gamma_{init}=1$).
>
> These results indicate that a zero initialization may lead to faster convergence, but it may also bias the model toward solutions that closely follow the enforced symmetries, potentially limiting exploration in the early stages of training. In contrast, initializing $\gamma_{init}=1$ results in slower convergence, but can lead to better final performance. Due to this year's NeurIPS policy restricting the use of images in the rebuttal, we are unable to include optimization dynamics plots at this stage. However, we will include these plots along with a detailed discussion in the Appendix of the final version.
>
> ---
>
> **Limitations Section:**
>
> For the final version, we will move the limitations discussion to a dedicated section before the conclusions.
>
> ---
>
> Once again, we would like to kindly thank the reviewer for taking their time to review our work and provide us with constructive feedback. We believe that addressing these points has helped us significantly improve the clarity of our paper and allowed us to add some important discussions to it.
>
> ---
>
> [27]: Wang et al., "Discovering Symmetry Breaking in Physical Systems with Relaxed Group Convolution", ICML 2024

---

> > ### Comment · Reviewer_k8VA · 2025-08-03
> >
> > Thank you for your rebuttal. I believe the paper is already quite good as it stands, but if the authors can include the 2D convolution experiment in the final version, it would be even more impressive. I still recommend accepting the paper.

---

### Official Review · Reviewer_ZRDi · 2025-06-30

**Clarity:** 3
**Significance:** 3
**Originality:** 3
**Rating:** 5
**Confidence:** 4

**Summary:**

This paper introduces a novel technique for training approximately equivariant networks. Specifically, the networks are assumed to have an architecture consisting of a linear combination of equivariant layers with non-equivariant layers, where the weights of the latter are learned through optimization to control the degree of non-equivariance. In particular, the authors' principal contribution is a specialized training method based off of the principles of homotropy via interpretation of the objective as a generalized Lagrangian.

**Questions:**

The key question I have is regarding the implementation. It seems like the application of Algorithm 2) would require a modification of a standard optmizer (such as Adam/AdamW).  Perhaps I missed it, but it appears that this is not discussed in the paper.

Is it necessary to modify the optimizer to handle the proposed training method? If not that is a big benefit and should be stated. If not, then a detailed explanation of a general recipe should be stated in the main paper or appendix, in addition to the release of the authors code.

**Ethical Concerns:**

["NO or VERY MINOR ethics concerns only"]

**Final Justification:**

I had very minor concerns coming into the rebuttal, which the authors fully addressed.

I think this is a strong paper about an important topic, and a necessary step for bridging the gap between equivariant theory and real-world performance.

I think this paper is a clear accept, and will be happy to advocate for it.

**Limitations:**

Yes.

**Paper Formatting Concerns:**

None.

**Quality:**

3

**Strengths And Weaknesses:**

I'm a fan of this paper. The authors contributions are well scoped, well explained, and supported by rigorous theoretical and extensive experimental results. In particular, the latter show broad applicability across a wide variety of tasks and architectures.   I think moving away from strictly equivariant models is important, as it is becoming clear that such models are ill-suited for practical applications where symmetries are often partial, messy, or otherwise corrupted.

Overall I think this paper will be a nice contribution to the conference and I am happy to recommend clear acceptance. That said, I have some minor questions/concerns below.

---

> ### Author Rebuttal · Authors · 2025-07-30
>
> We would like to thank the Reviewer for reviewing our work and for their positive assessment!
>
> In the following, we address their concern regarding the implementation of the primal-dual algorithm:
>
> For both the strictly equivariant version (Algorithm 1) and the partially equivariant Resilient Constrained Learning version (Algorithm 2), only minimal modifications are required. In practice, implementing a new optimizer is not necessary. In our codebase, we use the standard PyTorch implementations of SGD or Adam (though any optimizer can be used for either the primal or dual updates) and define separate optimizers for the primal and dual variables. The same approach can be easily adapted to other frameworks such as TensorFlow and JAX.
>
> ---
>
> In the following, we provide a practical PyTorch example showing how a standard equivariant architecture with a conventional optimization routine can be adapted to use ACE:
>
> 1. Modifications to the model - adding the $\gamma$ and the non-equivariant parameters:
>
>
> ```
> class Model(nn.Module):
>     def __init__(...):
>         (...)
>         self.gammas = nn.ParameterList([nn.Parameter(torch.tensor(1.)) for _ in range(n_layers)])
>         self.f_neq = nn.ModuleList([NoneqModule(...) for _ in range(n_layers)])
>         self.f_eq = nn.ModuleList([EqModule(...) for _ in range(n_layers)])
>
>     def forward(...):
>         (...)
>         for i in range(len(self.gammas)):
>             x = self.f_eq[i](x) + gamma[i] * self.f_neq[i](x)
>         (...)
>
> ```
>
>
>
> 2. Modifications to the training routine:
> ```
> gammas = model.get_gammas()
> dual_parameters = []
> lambdas = [torch.zeros(1, dtype=torch.float, requires_grad=False).squeeze() for _ in gammas]
> dual_parameters.extend(lambdas)
>
> # If we want to use Algorithm 2, add:
> # us = [nn.Parameter(torch.zeros(1, requires_grad=True).squeeze()) for _ in range(len(lambdas))]
> # dual_parameters.extend(us)
>
> optimizer_primal = optim.Adam/SGD/etc(model.parameters())
> optimizer_dual = optim.Adam/SGD/etc(dual_parameters)
>
> # Now, in the training routine:
> (...)
> loss = loss_fn(preds, y)
> dual_loss = 0
> slacks = [gamma for gamma in gammas]
>
> for dual_var, slack in zip(lambdas, slacks):
>     dual_loss += dual_var * slack
>
> for i, slack in enumerate(slacks): # Explicit, but can be obtained via autodiff
>     lambdas[i].grad = -slack
>
> # If we use Algorithm 2, change to:
> # loss = loss + 1/2 * torch.norm(us) ** 2
> # dual_loss = 0
> # slacks = [torch.norm(gamma) - u for gamma, u in zip(gammas, us)]
> # for dual_var, slack, u in zip(lambdas, slacks, us):
> #    dual_loss += (dual_var * slack) - (u * slack)
> # for i, slack, u in enumerate(zip(slacks, us)):
> #    lambdas[i].grad = -slack - u
>
> lagrangian = loss + dual_loss
> lagrangian.backward()
> optimizer.step()
> optimizer_dual.step()
>
> # If we use Algorithm 2, add:
> # for lambda in lambdas:
> #     lambda.clamp_(min=0)
>
> ```
>
> We will include this example on implementing ACE in the Supplementary material of the final version of the paper, along with links to the official implementation.
>
> ---
>
> We would once again thank the Reviewer for their feedback and are open to address any further questions or concerns that they might have!

---

> > ### Comment · Reviewer_ZRDi · 2025-08-03
> >
> > Thank you for the clarification and for the psuedocode.
> >
> > I am quite happy with this paper, and I think it is a clear accept that will make a great addition to the conference.

---

### Official Review · Reviewer_hCpi · 2025-07-03

**Clarity:** 4
**Significance:** 4
**Originality:** 4
**Rating:** 6
**Confidence:** 5

**Summary:**

It is well known that equivariant models cause complex loss functions even if the data obey symmetry and lower the expressiveness of the model. The authors propose Adaptive Constrained Equivariance: an optimization framework that starts from a non-equivariant model and introduces gradually equivariance via homotopy enabling thus a smooth transition from a non-equivariant to an equivariant model.

**Questions:**

Q1: L. 214 Not clear what it means by bounded translation.

Q2: It is not clear what symmetry was assumed in the MoCap dataset. What is the homogeneous space and which is the transitive action? Is it a global SO(3)?

Q3: While the gamma and loss plots give some insight, I think a very simple example with two weights and a very easy symmetry like SO(2) could give a very clear insight in the studied behaviors.

**Ethical Concerns:**

["NO or VERY MINOR ethics concerns only"]

**Final Justification:**

One of the best papers I have reviewed for Neurips in the last few years. The responses to all reviewers are illumnating.

**Limitations:**

none identified.

**Paper Formatting Concerns:**

none.

**Quality:**

4

**Strengths And Weaknesses:**

S1: The first ever optimization that automatically transitions from a free to an equivariant model.

S2: Theoretical guarantees on the equivariance approximation.

S3: It avoids manual tuning of losses and optimization schedules

S4: The authors cleverly exploited the fact that dual methods in constrained optimization are instances of homotopy continuation. This is explained well by starting with the dual of the pure equivariant problem, namely the empirical Lagrangian, which approximates the solution of the equivariant problem. Algorithms 1 and 2 juxtaposed illuminates very well the difference.

S5: Stopping the iteration over the non-equiv contribution $\gamma$ is formally shown to produce a bounded equivariance error if both equivariant and non-equivariant models are Lipschitz continuous.

S6: The authors moreover show that when data are partially summetric the non-equivariant model approximately satisfies the equivariance constraint with a proven bound.

S7: The trade-off between performance and equivariance can be resolved with a resilient constrain loss based on slacks.

S8: It is awesome that the authors formalize three questions/hypotheses for the experiments.

---

> ### Author Rebuttal · Authors · 2025-07-30
>
> We're very happy to hear that Reviewer hCpi found our paper interesting and compelling!
>
> In the following, we will address the questions raised by the Reviewer:
>
> **Q1 - Bounded Translation clarification**: We did not find a reference to “translation” or “bounded” on line 214 and therefore assume that the Reviewer refers to line 204. There, we highlight that in most cases, the transformations that we are interested in are isometries, i.e., norm-preserving transformations (B = 1), the most common of which are rotations and translations (involving either padding or wrapping). The results of Theorem 4.2, however, hold for arbitrary B-bounded transformations. We will add additional clarifications regarding this.
>
> We might, however, have misunderstood the question of the Reviewer; if our answer is not satisfactory, we kindly ask the Reviewer to ask us for further clarification.
>
> ---
>
> **Q2 - Symmetry assumed in MoCap**: The symmetry assumed for MoCap is the Special Euclidean group $SE(3)$ --- so translations $\mu \in \mathbb{R}^3$ and rotation matrices $R\in SO(3)$. Intuitively, as mentioned in the original EGNO paper [1], "*As a practical example, to model dynamics by learning a function to predict $\mathcal{G}^{(t+1)}$ from $\mathcal{G}^{(t)}$, we hope that any rotations and translations applied to the current state coordinate $x(t)$ should be accordingly applied on the predicted next one $x(t+1)$, while the velocities $v$ should also rotate in the same way but remain unaffected by translations*". We will add further clarifications in the final version of our paper.
>
> ---
>
> **Q3 - Simple example with an easy symmetry**: We thank the Reviewer for this suggestion. As also noted by Reviewer k8VA, incorporating experiments with 2D equivariant convolutions would be a valuable addition to our paper.
>
> We are currently preparing experiments inspired by the Relaxed Group Convolutions introduced in Wang et al., 2024 [2], where the authors visualize the weights of a relaxed $C_4$-equivariant convolution under different symmetry groups, such as $C_2$. In a similar spirit, we plan to train a $C_4$-equivariant network using ACE and analyze how the non-equivariant component $f_\theta^{neq}$ changes across different symmetry settings.
>
> Furthermore, since $f_\theta^{neq}$ can be interpreted as a correction term to $f_\theta^{eq}$ in the presence of symmetry mismatch, we also plan to visualize and compare the contributions of $f_\theta^{eq}(x)$ and $\gamma f_\theta^{neq}(x)$ to the final prediction. These visualizations will help illustrate how ACE adapts when exact symmetry is not present.
>
> ---
>
> Once again, we would like to kindly thank the Reviewer for their overwhelmingly positive assessment of our paper and for their feedback. If they have any further questions, we would be happy to address them!
>
> ---
>
> [1]:  Xu et al., "Equivariant Graph Neural Operator for Modeling 3D Dynamics", ICML 2024
>
> [2]: Wang et al., "Discovering Symmetry Breaking in Physical Systems with Relaxed Group Convolution", ICML 2024

---

### Official Review · Reviewer_bnrd · 2025-07-05

**Clarity:** 4
**Significance:** 3
**Originality:** 3
**Rating:** 5
**Confidence:** 4

**Summary:**

This paper proposes a novel training framework that overcomes the limitations of training strictly equivariant models by treating equivariance as a soft constraint and utilizing techniques from the constraint optimization literature. Specifically, the proposed Adaptive Constraint Equivariant (ACE) framework augments the equivariant model with per-layer non-equivariant unconstrained layers for which the output is modulated by parameters $\gamma$. The problem of training an equivariant network is then expressed as a constraint optimization problem, where the equivariant constraint is reduced to the equality constraint of $\gamma=0$. The authors then propose to optimize the original problem by optimizing the dual problem via gradient descent on the model parameters and gradient ascent on the dual variables. Additionally, the authors provide supporting theoretical analysis of the approximation error that occurs when the network returns to the equivariant parametrization and the end of training, by setting $\gamma=0$, and describe a mechanism that can allow users to infer possible symmetry misspecification by looking at the convergence of the $\gamma$ variable. Finally, the experimental section provides empirical evidence demonstrating the benefits of the proposed ACE framework for a wide range of equivariant tasks.

**Questions:**

- What is the sensitivity of the optimization of the dual problem to the choice of learning rate for both the parameters of the model and the dual variables?
- Is it possible that the non-equivariant component of the homotopic architecture increases the scale of its outputs during training as a way to counteract the reductions of the $\gamma$ variable?
- What is the reason for the gap between the reported performance of VN-DGCNN in this paper and the performance reported in the original Vector Neurons paper?

**Ethical Concerns:**

["NO or VERY MINOR ethics concerns only"]

**Final Justification:**

In the rebuttal the authors addressed all of the concerns I expressed in my initial review. I believe that this work both provides a useful methodology for training equivariant and approximate equivariant method while also provides useful insights about the nature of the problem. Due to the above I raise my rating to 5.

**Limitations:**

Yes

**Paper Formatting Concerns:**

There is not formatting concerns for this paper

**Quality:**

3

**Strengths And Weaknesses:**

Strengths:
- The authors propose a task-agnostic framework that, in contrast with previous works, doesn't rely on a user-specified learning schedule and allows for a data-dependent modulation of the equivariant constraint during training.
- The framework can be applied to both equivarariant and approximate equivariant tasks.  Section 4.2 provides a nice heuristic for determining what symmetries are part of the tasks and whether a fully equivariant or approximately equivariant approach is more beneficial.
- The formulation of the homotopic architectures requires minimal assumptions about the type of model to which the ACE framework can be applied.

Weaknesses:
- While the authors claim that their framework doesn't require additional user tuning, the optimization of the dual problem introduces additional hyperparameters (e.g. the learning rate for the dual variable $\lambda$). There is no study of the sensitivity of the optimization to these additional parameters.
- There is a gap between the accuracy reported for VN-DGCNN in Figure 7 and the accuracy reported in the original paper for ModelNet40. Additionally, in Figure 7, there is a small typo in the name of the model used.
- In the homotopic architecture, the non-equivariant component is not constrained, and although it is a bounded operator by B, B is free to become arbitrarily large during training to counterbalance the reduction of $\gamma$. This process can increase the approximation error shown in Theorem 4.1, even from small $\gamma$ values.
- Minor: There is an inconsistency in the naming of the different methods between Figure 2 and Figure 7. In Figure 2, the normal SEGNN (which is an equivariant model) is referred to as "No Constraints" model, while in Figure 7, the equivariant VN-DGCNN is referred to as "Equivariant" model.

---

> ### Author Rebuttal · Authors · 2025-07-30
>
> We would like to thank Reviewer bnrd for their thoughtful and constructive review. In what follows we clarify the Reviewer's concerns.
>
> **W1 - Learning rate $\eta_d$ for the dual variables**: Both Reviewer bnrd and Reviewer bUHD point out that, while our method indeed doesn't require hand-crafted schedules for $\gamma$ or tuning of regularization hyperparameters, the dual problem optimization involves a new learning rate $\eta_d$. Empirically, we have observed that having a dual learning rate $\eta_d \approx \eta_p$ works well in most situations. Our recommendation from a practical perspective is to find a good learning rate $\eta_p$ and then set $\eta_d = \eta_p$, or to set $\eta_d$ to a slightly lower value (e.g. if $\eta_p = 9e^{-4}$ then set $\eta_d=8e^{-4}$). We usually had a harder time searching for $\eta_p$ than $\eta_d$, as can be seen in Supplementary Table 4. Using the default learning rates for $\eta_p$ (e.g., from the official implementation of a model) will generally produce good results, but we observed that in some cases they were suboptimal, as stated on Line 951 in the Supplementary.
>
> We have run a sensitivity analysis on the N-Body dataset (1000 samples). Please note that the table contains a baseline experiment ($\eta_d$=N/A means ACE is not used). The initial values for $\eta_p, \eta_d$ were obtained via the initial learning rate grid search (Table 4 of the Supplementary). We set $\eta_p=9e^{-4}$ for all experiments and only evaluate with the strictly equivariant projection $f_{eq}$.
>
> | $\eta_d$  | Test MSE ($\times 10^{-3}$) |
> | --------- | --------------------------- |
> | N/A       | $6.64\pm0.14$               |
> | $1e^{-5}$ | $6.10\pm0.25$               |
> | $8e^{-5}$ | $5.47\pm0.29$               |
> | $1e^{-4}$ | $5.24\pm0.29$               |
> | $8e^{-4}$ | $4.97\pm0.01$               |
> | $1e^{-3}$ | $5.03\pm0.09$               |
> | $8e^{-3}$ | $5.11\pm0.16$               |
>
> As can be seen, different learning rates $\eta_d$ can affect final performance. However, in all experiments, using ACE produced better results than the base model (N/A), with the best results being obtained when $\eta_p \approx \eta_d$.
>
> We agree with the reviewer that this is an important point. We believe that this discussion adds further insight to our method and is very useful from a practitioner's perspective. We will include the discussion together with additional plots in the next iteration of our paper.
>
> ---
>
> **W2 - Performance gap between official VN-DGCNN and our method**: In our experiments, we adopt the setup from Pertigkiozoglou et al. [1], a related work that also focuses on training equivariant models. This setup uses a sub-sampled resolution of 300 points per point cloud, in contrast to the original VN-DGCNN paper [2], which uses 1024 points. Additionally, in our implementation, we only tuned the primal and dual learning rates $\eta_p, \eta_d$, without modifying other components or performing a broader hyperparameter search.
>
> Due to these differences, both in input resolution and in the level of tuning, our results are not directly comparable to those reported in [2]. We will continue to explore the source of these differences and include the additional results and a discussion in the final version of our paper.
>
> What is more, we thank the Reviewer for pointing out the typo in Figure 7.
>
> ---
>
> **W3 - Having a large norm for the non-equivariant component might increase the scale of its outputs:** While it is indeed true that the magnitude of the output of the non-equivariant layer $f_\theta^{neq,i}$ might dominate the constraint $\gamma_i$, we assume that the layer is M-Lipschitz and a bounded operator with $||f_\theta^{neq,i}||\leq B||x||$ holding throughout training. This can be easily enforced by reparametrizing the weights, e.g., by using Spectral Normalization [3] (L180-181) so that $B\leq 1$.
>
> From a practical standpoint, when seeking to obtain exact equivariance (Algorithm 1), we did not need to use Spectral Normalization --- when doing model selection, we checkpoint based on the validation performance of the equivariant projection $f_\theta^{eq}$, ignoring the non-equivariant component. However, we have observed that in the case of Resilient Constrained Learning (Algorithm 2), Spectral Normalization is needed for the equivariance error of $f_\theta^{eq} + f_\theta^{neq}$ to decrease (Figure 4); this is indeed due to B not being bounded if the weights are not reparametrized.
>
> We will add a more explicit discussion regarding the use of Spectral Normalization in the final version of our paper.
>
> ---
>
> **W4 - Naming Inconsistencies:** We thank the Reviewer for pointing out the naming inconsistencies. We will fix them for the final version.
>
> ---
>
> Once again, we kindly thank the Reviewer for taking their time to review our paper. We believe that addressing their concerns has helped us significantly improve the clarity of our paper and allowed us to add some interesting discussions to it. We are open to further discussions in the next reviewing phase!
>
> ---
>
> [1]: Pertigkiozoglou et al., "Improving Equivariant Model Training via Constraint Relaxation", NeurIPS 2024
>
> [2]: Deng et al, "Vector Neurons: A General Framework for SO(3)-Equivariant Networks", ICCV 2021
>
> [3]: Miyato et al, "Spectral Normalization for Generative Adversarial Networks", ICLR 2018

---

> ### Comment · Reviewer_bnrd · 2025-08-07
> **Response to Rebuttal**
>
> I thank the authors for their rebuttal and for addressing my concerns.
> I appreciate the inclusion of additional discussion regarding the choice of hyperparameters used in this work since it can significantly increase the ease of applying this method in different models and tasks.
> I thank the authors for clarifying my misunderstanding regarding the different setup between this work and [2].
> Finally, regarding the use of Spectral Normalization, I believe that its use is  important to be more extensively discussed in the paper, since (relating to my initial concern) it can also add additional hyperparameters that are important for the method.
> Since most of my concerns are addressed by the authors who are willing to add the additional discussions in the update version of the paper, I will increase my score to 5.

---

> > ### Author Response · Authors · 2025-08-07
> >
> > We would once again like to thank the Reviewer for their valuable suggestions and their positive reassessment of our work.
> >
> > We believe that addressing the concerns raised by the Reviewers has helped us to significantly improve both the clarity and completeness of our paper.

---

### Note · Authors · 2025-08-11

We thank the Reviewers for their constructive feedback and for the encouraging and positive evaluation of our work. Their comments have helped us further improve the paper, and we believe the revisions described below will reinforce this positive assessment.

Reviewers **bUHD** and **bnrd**:
1. **Dual learning rate ($\eta_d$):**
	- Add detailed discussion on the $\eta_d$–$\eta_p$ relationship, supported by new sensitivity analysis results.
	- Provide practical guidelines for setting $\eta_d$ (to be included in the Supplementary).
2. **Assumption 1 and the non-equivariant component’s norm**:
	* Expand the discussion on potential scale issues from $f_\theta^{neq,i}$ and explain boundedness assumption ($M$-Lipschitz, $B \leq 1$).
	* Explicitly mention that Spectral Normalization can enforce boundedness and when it is necessary (not needed for Algorithm 1, needed for Algorithm 2).

Reviewers **k8VA** and **hCpi**:
1. **New experiments with 2D convolutions:**
	* Add the new experiments performed on the $C_4$/$C_2$ data similar to Wang et al., where we visualize how the $f_\theta^{neq}$ weights change when symmetries are broken, and how $f_\theta^{neq}(x)$ can act as a correction term.

Reviewer **bnrd**:
1. **Performance gap when compared to the official VN-DGCNN:**
	* Clarify use of Pertigkiozoglou et al. setup and discuss differences from Deng et al. in the Supplementary.

Reviewer **hCpi**:
1. **Clarifications for line 214:**
	* Add further clarifications that in most cases, the transformations that we are interested in are isometries.
2. **Clarifications regarding assumed symmetries**:
	* Explicitly state what symmetries are assumed for the datasets (e.g. SO(3) for CMU MoCap).

Reviewer **ZRDi**:
1. **Code for optimization routine:**
	* Add an example implementation to the Supplementary.
	* **We will make the code for all of our experiments publicly available.**

Reviewer **k8VA**:
1. **Text improvements**:
	* Add one extra content page, reorganize into subsections, expand descriptions of setups, and provide more detailed result analysis.
	* Specify for each experiment which symmetries are exact vs. approximate.
	* Move the limitations discussion to a dedicated section before the conclusions.
2. **Initializing $\gamma$ at different values:**
	* Add discussion and results comparing $\gamma_{init} = 1$ vs. $\gamma_{init} = 0$ on the N-Body dataset.

**General**:
* Correct all reported typos.

---

### Decision · Program_Chairs · 2025-09-17

**Decision:**

Accept (oral)

**Comment:**

The paper presents a novel method for optimizing equivariant neural networks by initially utilizing a non-equivariant model and gradually transforming it to become increasingly equivariant. This approach cleverly applies homotopy methods from optimization. All the reviewers appreciated this innovative strategy and awarded high scores. I am pleased to recommend acceptance for an oral presentation.